# Independent amylase gene copy number bursts correlate with dietary preferences in mammals

Petar Pajic[1,2], Pavlos Pavlidis[3], Kirsten Dean[1], Lubov Neznanova[2], Rose-Anne Romano[2], Danielle Garneau[4], Erin Daugherity[5], Anja Globig[6], Stefan Ruhl[2]*, Omer Gokcumen[1]*

[1]Department of Biological Sciences, University at Buffalo, The State University of New York, New York, United States; [2]Department of Oral Biology, School of Dental Medicine, University at Buffalo, The State University of New York, New York, United States; [3]Institute of Computer Science (ICS), Foundation for Research and Technology – Hellas, Heraklion, Greece; [4]Center for Earth and Environmental Science, Plattsburgh State University, New York, United States; [5]Cornell Center for Animal Resources and Education, Cornell University, New York, United States; [6]Friedrich-Loeffler-Institut, Federal Research Institute for Animal Health, Greifswald, Germany

**Abstract** The amylase gene (*AMY*), which codes for a starch-digesting enzyme in animals, underwent several gene copy number gains in humans (Perry et al., 2007), dogs (Axelsson et al., 2013), and mice (Schibler et al., 1982), possibly along with increased starch consumption during the evolution of these species. Here, we present comprehensive evidence for *AMY* copy number expansions that independently occurred in several mammalian species which consume diets rich in starch. We also provide correlative evidence that *AMY* gene duplications may be an essential first step for amylase to be expressed in saliva. Our findings underscore the overall importance of gene copy number amplification as a flexible and fast evolutionary mechanism that can independently occur in different branches of the phylogeny.
DOI: https://doi.org/10.7554/eLife.44628.001

*For correspondence:
shruhl@buffalo.edu (SR);
gokcumen@gmail.com (OG)

**Competing interests:** The authors declare that no competing interests exist.

## Introduction

Diet has been a significant adaptive force in shaping human and nonhuman primate variation (*Hardy et al., 2015*; *Milton, 1981*; *Zhang et al., 2002*). One of the best-described examples of diet-related adaptation is the expansion of the copy number of the amylase gene in concordance with the increase of starch consumption in the human lineage (*Perry et al., 2007*), likely postdating the human Neanderthal split (*Inchley et al., 2016*). A gene duplication in the ancestor of Old World monkeys and great apes initially led to the formation of two amylase genes (*AMY2A* and *AMY2B*) with pancreas-specific expression (*Samuelson et al., 1990*). Then, a subsequent gene duplication in the ancestor of great apes led to the formation of *AMY1* which gained salivary gland-specific expression (*Meisler and Ting, 1993*). In the human lineage, further gene copy number gains of *AMY1* led to increased expression of the AMY1 enzyme in human saliva (*Perry et al., 2007*). Gene copy numbers of both *AMY1* and *AMY2* vary in different human populations (*Carpenter et al., 2015*; *Usher et al., 2015*), the former correlating with the extent of traditional starch consumption in these communities (*Perry et al., 2007*).

**eLife digest** Many mammals can digest starch by using an enzyme called amylase, but different species eat different amounts of starchy foods. Amylase is released by the pancreas, and in certain species such as humans, it is also created by the glands that produce saliva, allowing the enzyme to be present in the mouth. There, amylase can start to break down starch, releasing a sweet taste that helps the animal to detect starchy foods.

Curiously, humans have multiple copies of the gene that codes for the enzyme, but the exact number varies between people. Previous research has found that populations with more copies also eat more starch; if this correlation also existed in other species, it could help to understand how diets influence and shape genetic information. In addition, it is unclear how amylase came to be present in saliva, as the ancestors of mammals only produced the protein in the pancreas.

Pajic *et al.* analyzed the genomes of a range of mammals and found that the more starch a species had in its diet, the more amylase gene copies it harbored in its genome. In fact, unrelated mammals living in different habitats and eating different types of food have similar numbers of amylase gene copies if they have the same level of starch in their diet.

In addition, Pajic *et al.* discovered that animals such as mice, rats, pigs and dogs, which have lived in close contact with people for thousands of years, quickly adapted to the large amount of starch present in human food. In each of these species, a mechanism called gene duplication independently created new copies of the amylase gene. This could represent the first step towards some of these copies becoming active in the glands that release saliva.

In people, having fewer copies of the amylase gene could mean they have a higher risk for diabetes; this number is also tied to the composition of the collection of bacteria that live in the mouth and the gut. Understanding how the copy number of the amylase gene affects biology will help to grasp how it also affects health and wellbeing, in humans and in our four-legged companions.

DOI: https://doi.org/10.7554/eLife.44628.002

While the evolution of the amylase locus in the human lineage is well described, its evolution in other mammals is less well understood. Some studies have produced intriguing findings. For example, it was shown that mice, rats, and pigs express substantial levels of amylase in their saliva (*Boehlke et al., 2015*; *Janiak, 2016*) In addition, the amylase locus has been shown to be evolving under positive selection in dogs and in house mice (*Reiter et al., 2016*; *Staubach et al., 2012*). However, a comprehensive analysis of the evolutionary dynamics shaping the amylase locus across mammals is missing. Within this context, one interesting question is how amylase has evolved in animals who live in a commensal relationship with humans. Recent studies, for instance, have shown that dogs gained multiple copies of the amylase gene after their split from the wolf 30,000–40,000 years ago (*Skoglund et al., 2015*). This might likely be a result of their domestication, which exposed them to human food leftovers rich in starch (*Axelsson et al., 2013*; *Botigué et al., 2017*; *Ollivier et al., 2016*; *Reiter et al., 2016*). Thus, the evolution of amylase in other domesticated or human commensal mammals remains an alluring area of inquiry. Similarly, our understanding of the evolution of the amylase locus within the primate lineage remains limited. For instance, it is not known why some Old World monkeys have substantial amylase enzymatic activity in their saliva, despite missing the amylase duplication found in great apes (*Janiak, 2016*).

Here, we address three areas of inquiry with regard to the evolution of the amylase locus in mammals: (i) Can the link between diet and amylase evolution, suggested in the human lineage, be generalized to other mammals? (ii) What are the evolutionary forces that shape amylase copy numbers in mammals? (iii) What are the genetic mechanisms in different mammals leading to expression of amylase in salivary glands? To answer these questions, we pursued a comprehensive investigation of amylase gene copy numbers and salivary expression across multiple mammalian lineages.

## Results and discussion

### Amylase copy number gains occurred in multiple mammalian lineages independently

The human lineage-specific amylase gene duplications were initially thought to represent a unique case of evolutionary adaptation to increased starch consumption in humans (*Perry et al., 2007*). Up to five more haploid copies of the amylase gene can be found in humans than in chimpanzees. Therefore, the recent revelation that a similar increase in amylase gene copy number occurred in dogs (*Axelsson et al., 2013*; *Ollivier et al., 2016*) is remarkable since it shows that the same gene underwent bursts of gene copy number gains in two separate species independently. Copy number variation was also noted in pigs (*Paudel et al., 2013*). To comprehensively investigate amylase gene copy number gains in other mammalian lineages, we conducted a digital droplet polymerase chain reaction (ddPCR)-based analysis of amylase gene copy numbers from 204 DNA samples across 46 species encompassing all major branches of the mammalian phylogeny. In addition to humans and dogs, we discovered similar bursts (i.e. gains of more than one copy) of amylase gene copy number in mice, rats, pigs, and boars (*Figure 1—figure supplement 1*, *Supplementary file 1*).

We hypothesized that the elevated gene copy numbers observed in different branches of the mammalian phylogeny (*Figure 1*) result from independent duplication events. An alternative explanation would be that the ancestor of placental mammals had multiple copies of the amylase gene, which were subsequently lost in certain mammalian lineages. To distinguish between these two scenarios, we constructed from available reference genomes a maximum likelihood tree of amylase coding sequences (*Figure 2A*, see *Figure 2—figure supplement 1* for a more comprehensive tree with outgroups included). Our results showed that amylase genes within a given species are more similar to each other than they are to those of other species. One explanation for this observation could be that duplications of the amylase gene might have occurred in each lineage independently.

Yet another explanation could be that lineage-specific gene conversion events occurred among ancestral amylase copies. Indeed, inter- and intra-chromosomal crossover has been shown in the amylase locus of humans (*Groot et al., 1990*; *Gumucio et al., 1988*). Such a process, if it occurred frequently enough, could potentially generate high similarity among amylase gene copies in any given mammalian lineage. These two scenarios, independent lineage-specific duplication events and gene conversion among existing gene copies, are difficult to distinguish using phylogenetic analysis alone (*Mendes et al., 2018*). To solve this conundrum in humans, Samuelson *et al.* searched for lineage-specific signatures associated with individual gene copies in the amylase locus. They identified a retrotransposon (HERV_a_int) inserted upstream of a new amylase gene duplicate (*AMY1*) in the ancestor of great apes (*Samuelson et al., 1990*). This retrotransposon was found to be associated with all the additional *AMY1* copies detected in humans (*Perry et al., 2007*). This finding strongly supported the notion that the duplications of *AMY1* occurred after the human-chimpanzee phylogenetic split (*Perry et al., 2007*; *Samuelson et al., 1990*).

Based on that, we asked if similar retrotransposons or other genomic signatures could help us determine whether the amylase gene copy number bursts in other mammalian genomes occurred independently. We first interrogated the mouse reference genome, as it is adequately complete for such an analysis. Indeed, we found a mouse lineage-specific retrotransposon (L1Md_T) in the upstream region of five out of the seven mouse amylase gene copies. The presence of this retrotransposon along with the duplicated copies parallels the situation in humans (*Figure 2B*). By ddPCR analysis, we found 9–13 diploid copies of the amylase gene in brown rats and 6–7 copies in black rats and wood rats (*Supplementary file 1*). Considering the close phylogenetic relationship of rats and mice, we expected that the high copy number of amylase had evolved in their rodent ancestor. However, the L1Md_T retrotransposon was not found in rats. When we investigated the amylase locus in the rat genome, we found a rat-specific retrotransposon (L1_Rat3) inserted upstream of two out of the three rat amylase copies in the assembled rat reference genome (*Figure 2B*). Therefore, amylase gene duplications in rats likely occurred independently from the ones in mice. The amylase gene copy numbers in rats found by ddPCR do not match those annotated in the rat reference genome. This inconsistency could be due to major sequence gaps in the assembled amylase locus (indicated by line breaks in *Figure 2B*). As an additional complication, we discovered that one of the gene copies, *amy1* in mouse (discussed in the next section within the context of salivary expression, see *Figure 2—figure supplement 2*), is shared with other rodent species including rats.

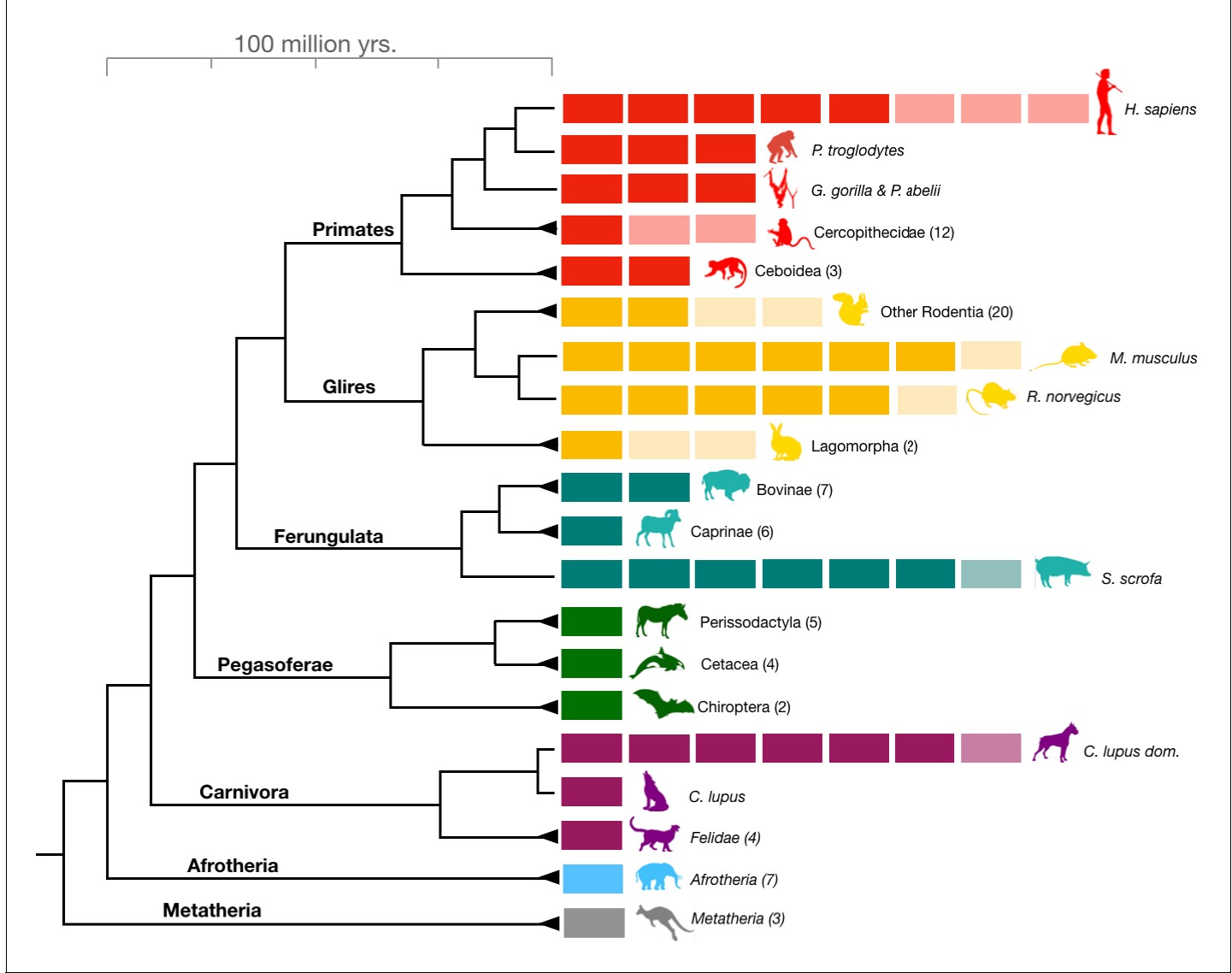

**Figure 1.** *Amylase* gene copy number bursts across mammals. Boxes represent all haploid amylase gene copies found in representative species or among clades across the mammalian phylogeny (see **Supplementary file 1** for a comprehensive dataset). Lighter colored boxes represent the variation in copy numbers found in at least two individuals of a given species or in reference genomes of at least two species within a clade. Triangles at the end of branches indicate that copy numbers of more than one species belonging to the same clade were shown together as a single column. The numbers in parentheses following clade names indicate the number of species used for estimating the gene copy numbers.

DOI: https://doi.org/10.7554/eLife.44628.003

The following figure supplement is available for figure 1:

**Figure supplement 1.** Primer design and ddPCR accuracy.

DOI: https://doi.org/10.7554/eLife.44628.004

Nevertheless, our results support that most amylase gene copy number gains occurred in mouse and rat lineages independently.

We also conducted similar analyses of the amylase locus in dogs and pigs (**Figure 2B**). Despite the fact that the assemblies of their genomes are not as complete as human and mouse reference genomes, we were able to investigate the genomic signatures in contigs that harbor distinct copies of amylase genes. In dogs, we found a canid-specific L1 element (L1_Canid) in all four amylase gene copies assembled across three different contigs. In pigs, we found an older lineage-specific L1 element (L1M3) downstream of all six amylase copies assembled across three contigs.

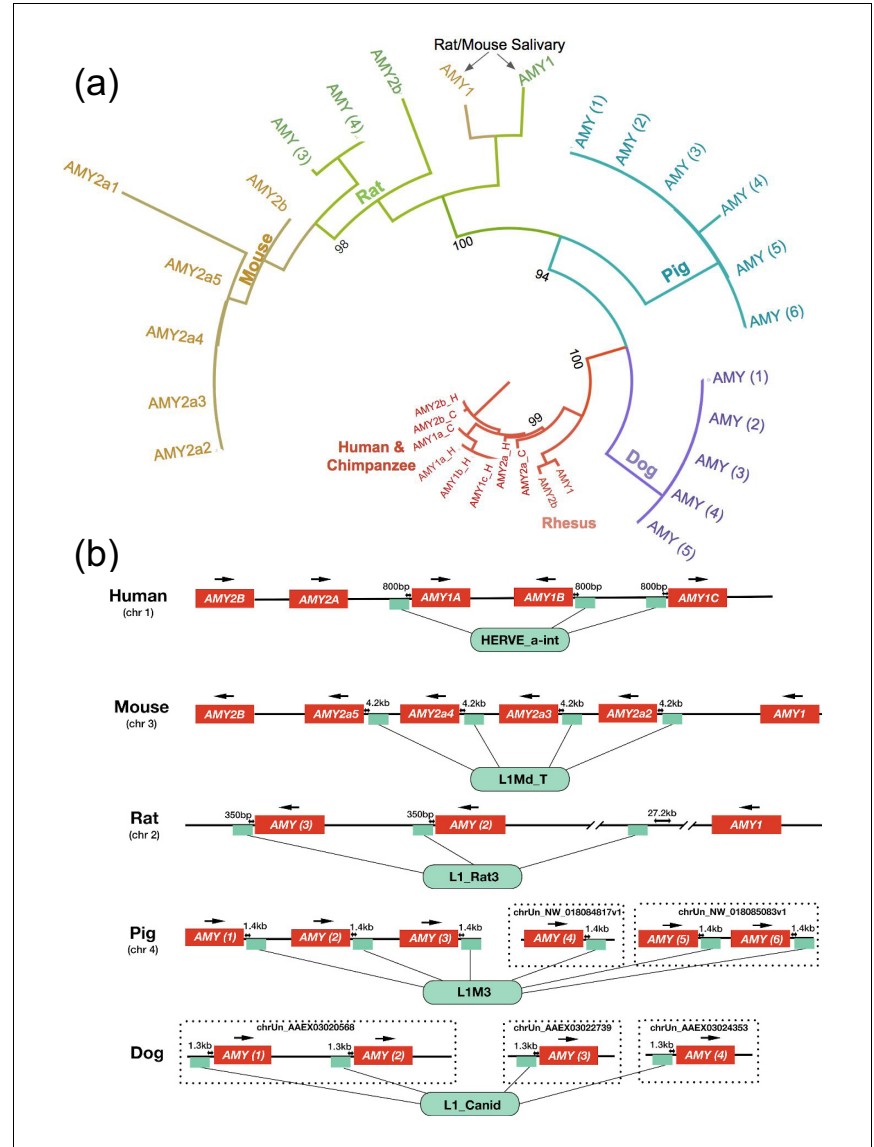

**Figure 2.** *Amylase* duplications evolved independently. (**a**) Maximum likelihood tree based on phylogenetic similarity of amino acid sequences of amylase gene copies translated from copies identified by BLAST. Bootstrap values are shown next to the major branch splits. A larger tree containing all bootstrap values and additional outgroup species can be found in *Figure 2—figure supplement 1*. (**b**) Types and locations of retrotransposons associated with amylase gene copies found in human, mouse, rat, dog and pig reference genomes. Small green boxes symbolize the positions of the retrotransposons. Arrows above individual amylase gene copies indicate the direction of transcription. Distances in kilobases between amylase gene copies and associated retrotransposons are shown above the green boxes. Non-assembled contigs are shown in dotted frames. Two major gaps around the third retrotransposon in the rat amylase locus are indicated as line breaks.

DOI: https://doi.org/10.7554/eLife.44628.005

The following figure supplements are available for figure 2:

**Figure supplement 1.** Expanded maximum likelihood amylase gene tree of mammalian reference genome sequences (PhyML).

DOI: https://doi.org/10.7554/eLife.44628.006

**Figure supplement 2.** RNA-sequencing data for expression of amylase genes in mouse parotid salivary gland.

DOI: https://doi.org/10.7554/eLife.44628.007

Overall, we found lineage-specific retrotransposons located in similar proximity (0.8–4.2 kb upstream or downstream) to multiple amylase gene copies in human, mouse, rat, pig, and dog reference genomes (*Figure 2B*). In each of these cases, the respective retrotransposons are in identical positions relative to the amylase gene within each species. We surmised that these retrotransposons inserted in proximity to an ancestral amylase copy in each species independently and were subsequently duplicated along with further gained amylase gene copies. These findings do not rule out that gene conversion as well as other mechanisms (incomplete lineage sorting, crossover events, and ancestral gene duplication polymorphisms) might have shaped variation in this locus. However, the fact that lineage-specific retrotransposons accompany amylase gene copy number gains in humans, mice, rats, dogs, and pigs, clearly points to lineage-specific duplications as a major driver of amylase gene copy number bursts in these species. The extent to which retrotransposons affect mutational or functional dynamics in the amylase locus remains an important area for future research.

## Expression of amylase in saliva evolved in different mammalian lineages independently and was facilitated by gene copy number duplication

Ancestrally, amylase was a pancreatic enzyme in mammals. However, in certain mammalian species, amylase became expressed also in saliva (*Chauncey et al., 1963*). In humans, this acquisition of salivary gland-specific expression has been well explained (*Ting et al., 1992*). It has been shown that the above-described retrotransposon insertion along with the *AMY1* duplicate in the ancestor of great apes was responsible for tissue-specific expression of this gene in salivary glands (*Samuelson et al., 1990*). Previous studies hypothesized that a similar, but independent gene duplication event led to the expression of amylase also in the saliva of mice (*Meisler and Ting, 1993*). It remained unresolved whether the mechanism that enabled the expression of amylase in mouse saliva is similar to that determined for humans. Moreover, even though some reports noted the expression of amylase in the saliva of various other mammalian species (*Janiak, 2016*), a comprehensive analysis of its expression across the mammalian phylogeny is still missing. Another unanswered question from an evolutionary perspective is what a potential adaptive benefit of expressing amylase in saliva could be. Even though amylase is a digestive enzyme, it is clear that in most mammals starch digestion primarily occurs through the activity of the pancreatic enzyme in the intestines rather than in the mouth (*Fernández and Wiley, 2017*).

To address these questions, we performed a screen across the mammalian phylogeny to investigate which lineages express amylase activity in saliva. We used a two-pronged approach, comprising a starch lysis plate assay (*Figure 3A*) and a high-sensitivity in-solution fluorescence-based amylase assay (*Figure 3B*). Currently, our study provides the most comprehensive documentation of salivary amylase activity in mammals, encompassing 127 saliva samples across 22 species (*Supplementary file 1*). This is a significant contribution given that previous studies varied considerably in sample preparation, methods of analysis, and assay sensitivity (*Janiak, 2016*).

Our results showed that amylase activity in saliva is more widespread among mammals than previously thought (*Figure 3B*). In addition to species that were already known to express amylase in their saliva, we observed salivary amylase activity in some New World monkeys, boars, dogs, deer mice, woodrats, and giant African pouched rats (*Supplementary file 1*). It is important to note here that our findings also suggest that amylase activity in dog saliva varies from breed to breed (*Supplementary file 1*). It remains to be determined to what degree this variable expression of amylase in the saliva of different dog breeds might have been the result of older adaptive forces, perhaps related to dogs becoming companions of humans thereby exposed to a human diet, rather than merely occurring as a byproduct of recent intentional breeding for other traits.

To explain the expression of amylase in the saliva of some mammalian lineages, but not others, we considered two scenarios. First, it is possible that salivary expression of amylase could be an ancestral trait that was subsequently lost in most species. Second, it is possible that salivary expression may have evolved independently in different lineages. To distinguish between these two scenarios, we asked whether orthologous copies of the amylase gene are expressed in humans and mice. Based on previous work showing that *AMY1* copies in humans are expressed in salivary glands, we asked which amylase copy is expressed in mouse salivary glands. By mapping parotid salivary gland RNA-Seq data (*Gluck et al., 2016*) to the mouse reference genome (mm9), we found that the copy annotated as mouse *amy1* is expressed in salivary glands, while the other amylase gene duplicates have a negligible expression in that tissue (*Figure 2—figure supplement 2*). However, mouse *amy1*

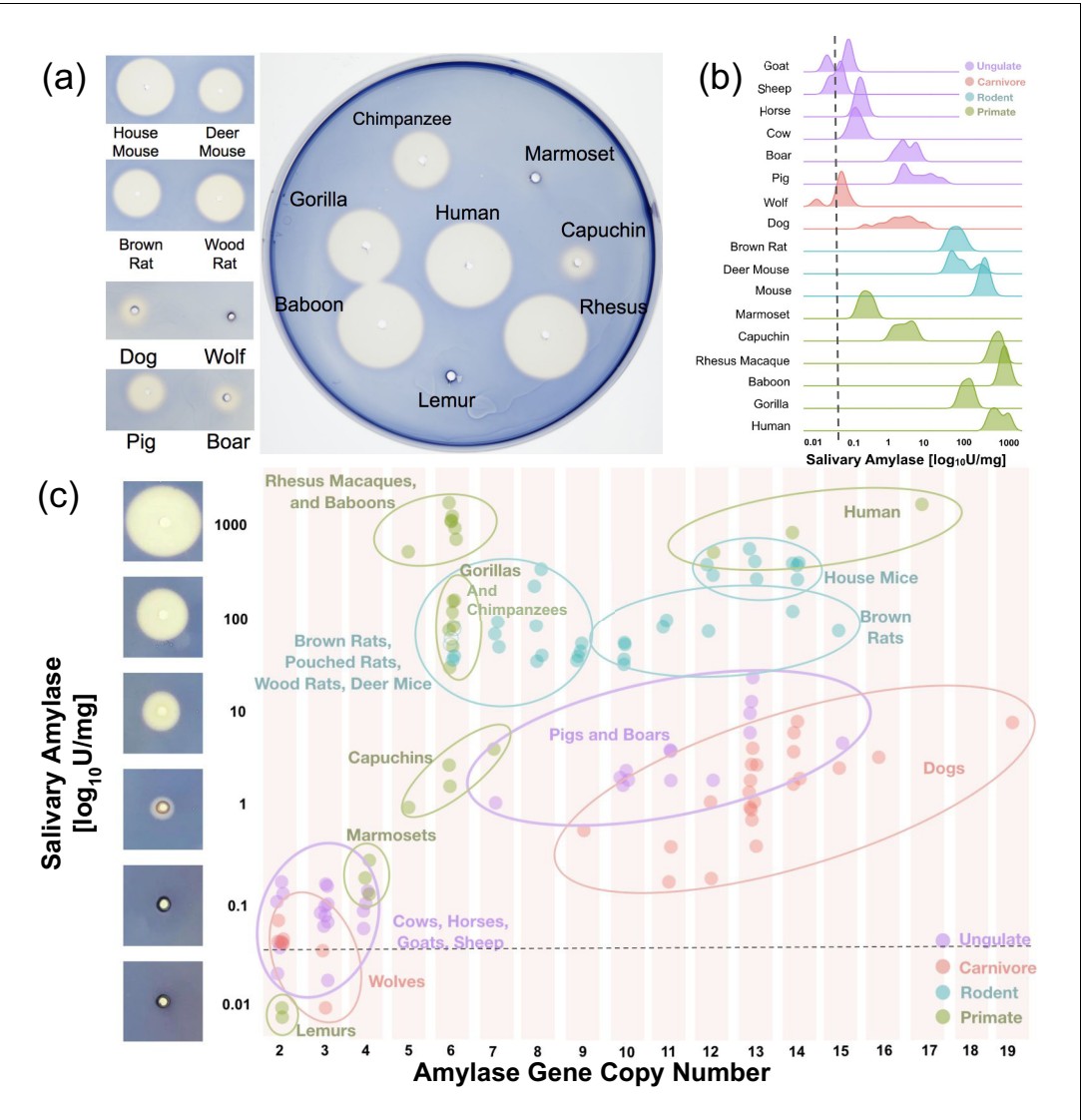

**Figure 3.** Salivary amylase activity and relationship to gene copy number. (**a**) Representative starch lysis plate assay showing the extent of lysis caused by the enzymatic activity of amylase in the saliva of various mammalian species. The left panel shows side-by-side comparisons of domesticated species and their counterparts in the wild. The agar plate shows the lysis caused by representative primate saliva samples. (**b**) Density plots showing salivary amylase activity in different species as measured by a high-sensitivity colorimetric assay. The dotted line represents the mean amylase activity level determined in the serum of humans, rats, pigs, boars, and gorilla (mean: 0.034 U/mg, range: 0.01–0.077 U/mg). A full dataset can be found in *Supplementary file 1*. (**c**) Scatter plot of amylase activity and gene copy number in multiple species as measured by ddPCR. The dotted line represents the same serum activity shown in (**b**). Images of starch plate lysis caused by standard dilutions of amylase are shown to the left of the y-axis next to their corresponding activity values.
DOI: https://doi.org/10.7554/eLife.44628.008

is not orthologous to human *AMY1* (*Meisler and Ting, 1993*). Furthermore, its amino acid sequence is distinct from those of other amylase copies in the mouse genome. This distinct *amy1* sequence is shared with rat and other rodents including deer mouse, vole, Mongolian gerbil, and golden hamster. This suggests that the duplication event that led to the formation of mouse *amy1*, the copy that is expressed in mouse salivary glands, likely occurred in an ancestor of *muroidea*. Interestingly, the more recently acquired, mouse and rat-specific amylase gene duplications are not expressed in mouse salivary glands (*Figure 2—figure supplement 2*).

We could not find a mammalian species that underwent a 'burst' of amylase gene copy number that did not show concurrent salivary amylase activity (*Figure 3C*). Also, we found no species with less than four diploid amylase copies that showed any measurable amylase activity in saliva. It is important to note here that the relationship between amylase gene copy number and salivary amylase activity cannot be explained by linear correlation. For example, rhesus macaques and baboons have relatively low amylase gene copy numbers (5–6 diploid copies) but show high amylase activity in their saliva (514–1,652 units per mg of total protein). In contrast, dogs, which among mammals showed some of the highest gene copy numbers of amylase (9–19 diploid copies), express very low amounts of amylase in their saliva (0–9 units/mg). Overall, these results suggest that a gene duplication may be a necessary condition for amylase to become expressed in saliva. Subsequently, however, different regulatory architectures are likely responsible for the differences in amylase activity observed in saliva across different mammalian species.

## A broad-range diet containing starch correlates with increased amylase gene copy number

For humans, it has been postulated that starch consumption has driven the increased gene copy number of the amylase gene through positive selection (*Perry et al., 2007*). For dogs, the rapid amylase gene copy number increase as compared to wolves (3–8 haploid copy number gain) has been associated with a transition during their domestication from a primarily meat-based diet to a diet enriched in starch (*Axelsson et al., 2013*). Based on these previous findings, we hypothesized that in other mammalian species, gains in gene copy number and the associated gain of amylase expression in saliva are likely driven by starch being a dietary component. Unfortunately, a systematic survey describing the amounts of starch consumption across mammals is lacking. Moreover, the amount of starch in the diet varies among subspecies, and sometimes even among geographically distinct populations of the same species (*Pineda-Munoz and Alroy, 2014*). Thus, to test the above hypotheses, we categorized mammals into those that consume specialized diets (strict carnivores or strict herbivores) and those consuming broad-range diets (including different amounts of starch). Based on extensive literature review (see Materials and methods), we subdivided mammals consuming broad-ranged diets further into those that consume diets containing high amounts of starch (humans, mice, brown and black rats, dogs, pigs, and boars) and those that consume low amounts of starch. We then conducted a comparative analysis of amylase copy number and salivary enzymatic activity among these categories (*Figures 4A, B and C*).

We found that species consuming a broad-range diet generally harbor significantly higher copy numbers of the amylase gene (*Figure 4A*). Specifically, the mean diploid amylase copy number among animals consuming specialized diets is 2.4, while it is 7.0 for those consuming broad-range diets. Corrected for phylogenetic dependence, a broad-range diet is significantly associated with increased amylase gene copy number ($p =\sim 9 \times 10^{-6}$). Among species that consume a broad-range diet, we found that those who over recent evolutionary time gained access to abundant starch-rich foods — either through domestication (as in the case of dogs and pigs) or through dietary commensalism with humans (as in the case of house mice or brown and black rats) — harbor considerably higher copy numbers of the amylase gene (*Figure 4A*). The mean diploid amylase gene copy number is 11.6 for species consuming a starch-rich diet, while it is 5.0 for those that consume lower amounts of starch in their diet. When corrected for phylogenetic dependence, starch consumption (categorized into high- and low-starch content in diet) is significantly correlated with amylase gene copy number ($p =\sim 10^{-4}$).

We found that mammals consuming a broad-range diet also express on average a thousand-fold higher amylase activity in their saliva (~243.6 units/mg of total salivary protein) than those consuming specialized diets (~0.1 units/mg) (*Figure 4B*). When corrected for phylogenetic dependence, it appeared that consumption of a broad-range diet is significantly associated with the enzymatic activity of amylase in saliva ($p =\sim 10^{-4}$). Salivary expression of amylase, however, turned out to not be associated with the amount of starch consumption per se (*Figure 4B*). Rather, salivary expression of amylase appeared to be associated with the consumption of starch, regardless of the amount. Previous work in humans and in rats showed that salivary amylase is linked to the perception of starch (*Mandel et al., 2010*; *Sclafani et al., 1987*). Based on this, one possible evolutionary explanation could be that the ability to enzymatically liberate sugar from long-chain starch molecules for the perception of sweet taste might have provided a metabolic fitness advantage to mammalian species

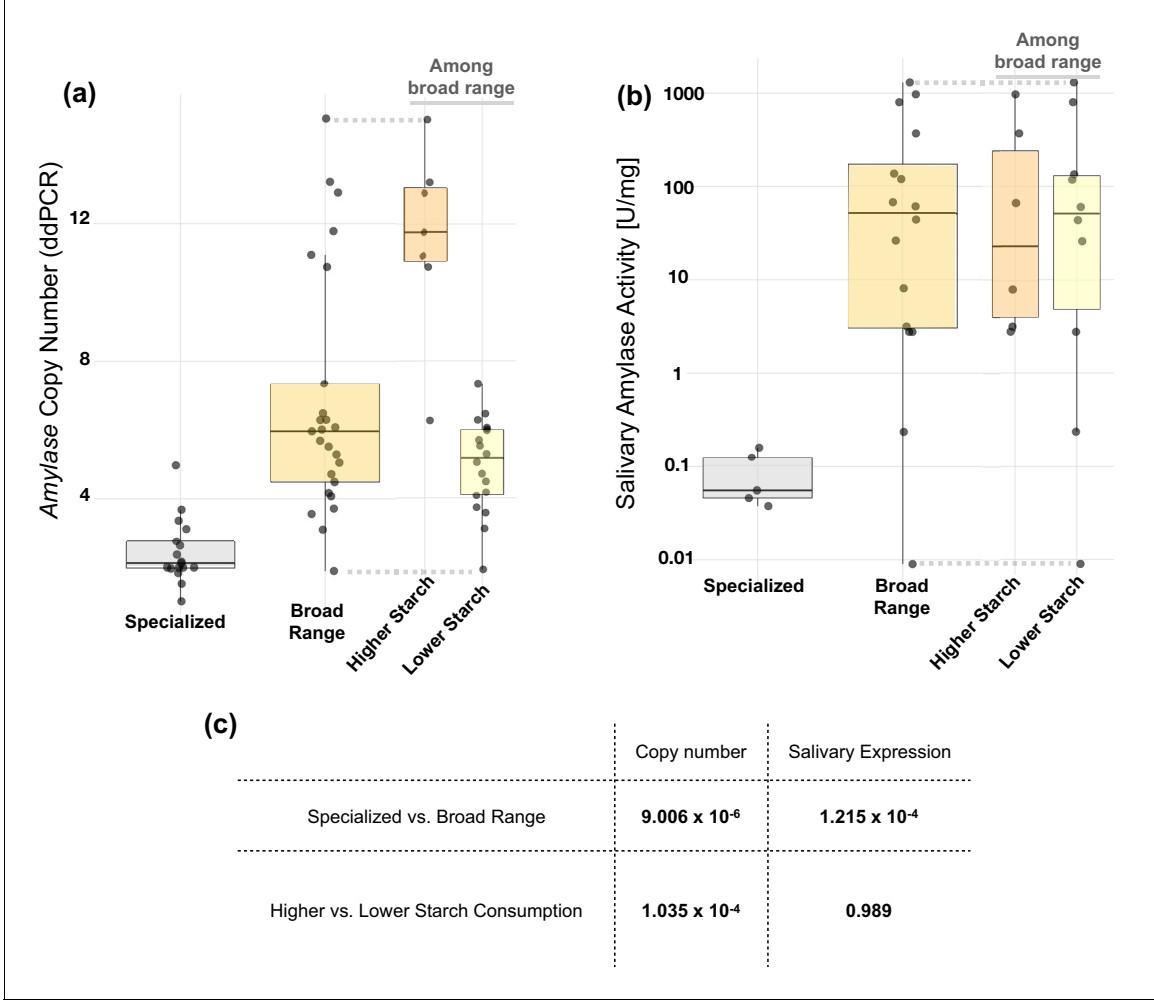

**Figure 4.** *Amylase* gene copy numbers and salivary enzyme activity in species with different dietary preferences. Box plots representing (**a**) *AMY* gene copy numbers and (**b**) salivary amylase activities (in units per milligram of total salivary protein) in mammalian species assigned by their dietary preferences into either specialized (carnivore or herbivore) or broad-range. The species consuming a broad-range diet were further sub-grouped into two categories based on the relative amounts of starch in their diet. (**c**) Estimates of statistical significances for the differences found between dietary groups. p-values were calculated for independent contrasts to account for phylogenetic confounding effects.
DOI: https://doi.org/10.7554/eLife.44628.009

consuming a broad range diet. We like to point to two outliers in our dataset of primate species consuming broad-range diets. We found that marmosets and lemurs express negligible amylase activity in their saliva (~0.2 and ~0.009 units/mg, respectively). However, for them starch is not a primary component in their otherwise diverse diet. Marmosets were shown to consume tree gum while lemurs consume tamarind fruit, reportedly rich in non-starch polysaccharides (for further information on starch consumption by mammalian species see Materials and methods).

Next, to investigate the influence of human commensalism, we conducted a comparative investigation of amylase gene copy number and enzymatic activity in saliva between mammalian species interacting with humans and their closest evolutionary relatives in the wild. In dogs, that due to their commensalism with humans consume a considerable amount of starch, we found an increase, not only in amylase gene copy number (*Axelsson et al., 2013*) but also in enzymatic activity of amylase in saliva as compared to the carnivorous wolf from which dogs diverged approximately 30,000–40,000 years ago (*Skoglund et al., 2015*) (*Figure 3A*). A less substantial increase was found in species that already consumed starch in their ancestral state (e.g. mice and rats which diverged from granivorous ancestors). Along the same lines, we found no difference of amylase gene copy numbers and salivary enzymatic activity between domesticated pigs and wild boars. This could be explained

because boars, the ancestral species, already consumed starch in amounts comparable to those of pigs, their domestic counterparts. In fact, previous observations showed that boars and humans have similar starch-rich ancestral diets due to their consumption of underground starch-containing storage stem tissues known as tubers (*Hatley and Kappelman, 1980*).

## Evolution of amylase in primates

In primates, we could conduct a finer resolution analysis of amylase evolution and its relationship to diet than in non-primate mammals because for primates we have access to more data and samples. Specifically, we could investigate the amylase gene locus in 14 different primate species (53 DNA samples) for gene copy number, and from 8 of these species, we also could obtain saliva (26 saliva samples) for measuring enzymatic activity (*Figure 5*). We confirmed previous studies which documented a duplication of the amylase gene in the ancestral population of the catarrhini and an additional duplication in the ancestral population of the great apes (*Meisler and Ting, 1993*). Among Old World monkeys, we found further amylase gene copies in rhesus macaques, baboons, and vervets, all species which consume a broad-range diet. In contrast, we found that leaf-eating Old World

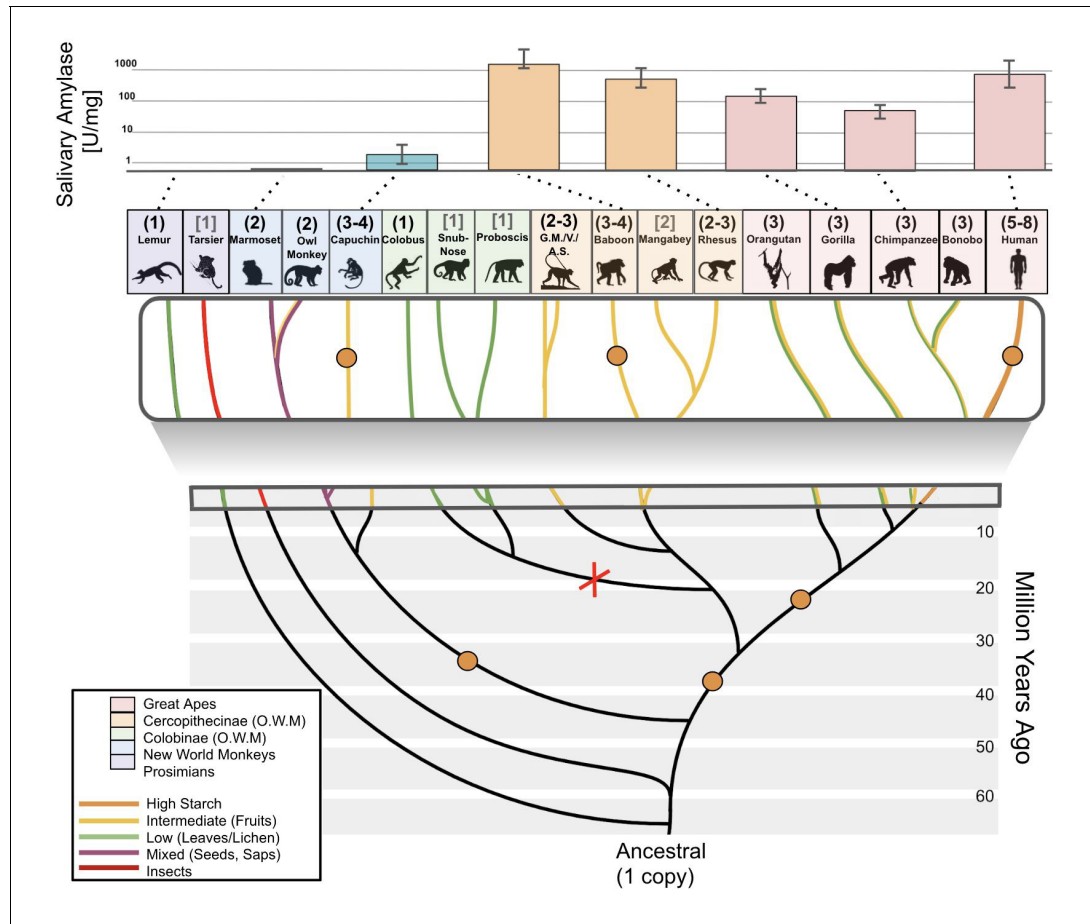

**Figure 5.** Amylase gene copy number duplication events and salivary activities across the primate phylogeny. Bars in the upper part of the diagram represent mean amylase activities in saliva of corresponding primate species (indicated by the dotted lines). Orange dots in the branches of the phylogenetic tree show the presumed occurrences of gene duplication events. The red X indicates a likely gene loss event. Haploid amylase gene copy numbers are indicated in parentheses above the species name. The copy numbers determined through genotyping by ddPCR are shown in black round parentheses, while those based on reference genomes are shown in gray square brackets. Phylogenetic branches are colored according to dietary preferences (see boxed legend). The upper boxed section is a zoomed-in version of the tree. Abbreviations: G.M., green monkey; V, vervet; A. S., Allen's swamp monkey; O.W.M., Old World monkeys.
DOI: https://doi.org/10.7554/eLife.44628.010

monkeys (colobus, snub-nose, and proboscis monkeys) (*Hohmann, 2009*) possess only one copy of the amylase gene, indicating a potential loss of a copy in this lineage.

Most New World monkey genomes that we tested carry two haploid amylase copies. Assuming that the ancestral state of this lineage had one haploid copy, our results suggest yet another occurrence of gene copy number gain in the ancestor of New World monkeys. Moreover, we found an additional amylase gene copy in capuchins, which generally consume more starch than other New World monkeys (*Galetti and Pedroni, 1994*; *Rowe and Myers, 2016*). Next, we investigated lemurs, an outgroup primate species to monkeys and great apes, and found that they indeed only harbor one haploid copy of the amylase gene (*Figure 5*). Our result in the lemur lineage, combined with prior reports that ancestors of simians have a single amylase gene copy (*Samuelson et al., 1996*; *Samuelson et al., 1990*), suggests that primate ancestors possessed only one haploid copy of the amylase gene. Still, other scenarios, involving multiple amylase copies in the primate ancestor followed by gene loss, may also explain the observed copy number variation across the primate phylogeny, and cannot be dismissed at the present stage, although they are more complex in their assumptions. Clearly, more studies are needed to fully resolve the evolutionary history of the variation in this locus in primates, which, in addition to gene duplications, may have been shaped by gene conversion and complex rearrangements as well as incomplete lineage sorting.

Next, we investigated whether variation in amylase gene copy numbers among primates translates into salivary expression in a similar way as we had shown for non-primate mammals. We found that several species of Old World monkeys, including rhesus macaques and baboons, express high enzymatic activity of amylase in their saliva (*Figure 5*) (*Mau et al., 2010*). These primates, belonging to the subfamily *cercopithecinae*, are known for their cheek pouches in which they store food for prolonged oral predigestion (*Lambert, 2017*; *Rahaman et al., 1975*). Thus, this primate subfamily could be an exception in that salivary amylase may substantially participate in oral digestion of starch.

Most New World monkeys do not consume starch in their regular diets. For example, marmosets primarily consume insects and plant exudate (*Rylands and Faria, 1993*), while owl monkeys consume flowers, insects, nectar, and leaves (*Rowe and Myers, 2016*; *Wright, 1994*). In agreement with their dietary habits, we found little to no salivary activity of amylase in these New World monkeys. Capuchin monkeys are an exception because they consume fruits, bulbs, and seeds (*Galetti and Pedroni, 1994*; *Rowe and Myers, 2016*). Accordingly, we discovered enzymatic activity levels of salivary amylase in capuchins that reach levels found in pigs and boars (*Figures 3C* and *5*).

Our results in primates document two additional instances (*cercopithecinae* and capuchins) where duplications of the amylase gene coincide with salivary expression. Combined, our results suggest that the evolution of the amylase locus in primates followed the same general trends observed for all mammals in that dietary strategies coincide both with amylase gene copy number and salivary expression in a lineage-specific manner.

## Conclusion and outlook: a working model to explain how the amylase locus evolved

Our results reveal a staggering diversity of amylase gene copy numbers across extant mammals that consume starch. We report multiple bursts of amylase copy number gains that occurred independently in different branches of the mammalian phylogeny. Our results showed that each of these bursts coincided with expression of amylase in saliva. Our results also showed that phylogenetically distant species living in different habitats and consuming different diets have arrived at astonishingly similar amylase gene copy numbers, which correlate with the level of starch in their diet. Building on earlier models of the locus' evolution (*Axelsson et al., 2013*; *Perry et al., 2007*; *Samuelson et al., 1990*) and using our own data, we deduce a model of how the amylase gene locus might have evolved across mammals (*Figure 6*). We posit here that the amylase locus evolved under the influence of the dietary context driven by the functional importance of amylase enzymatic activity in two digestive gland systems, namely the salivary glands, located at the entrance to the gastrointestinal tract, and the pancreas located further distally in the digestive continuum.

Most evolutionary models agree that the ancestral mammalian amylase gene was expressed in the pancreas. It has been suggested that increase in gene copy number leads to higher amylase expression in the pancreas, which in turn allows rapid and effective intestinal digestion of starch in species consuming a higher amount of that food component in their diet (*Axelsson et al., 2013*).

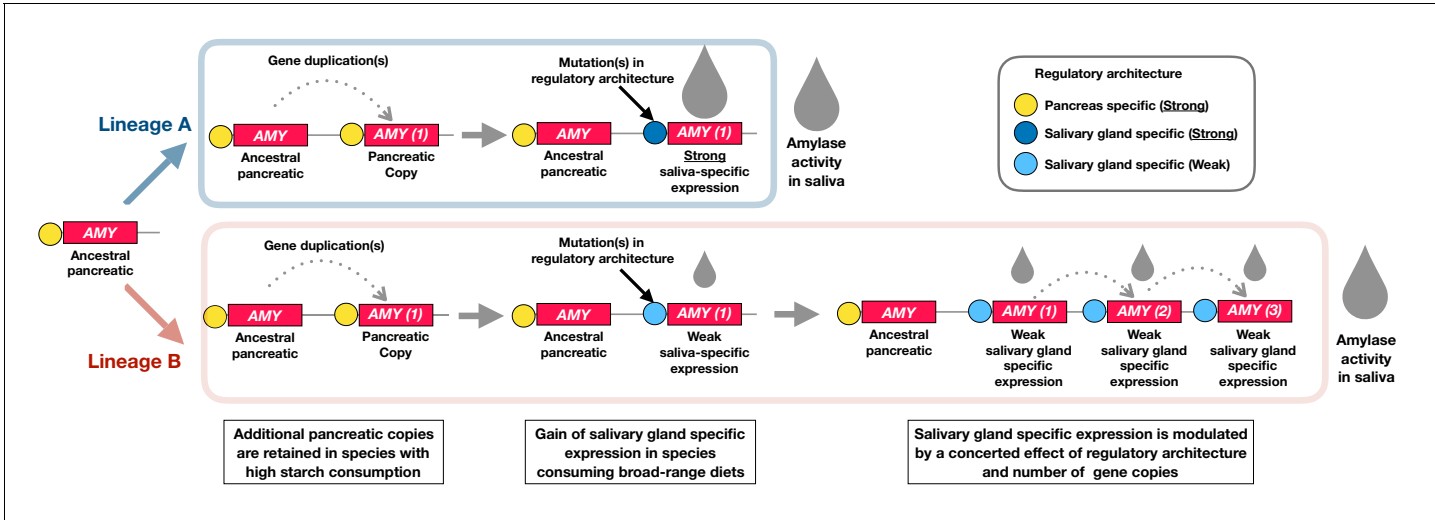

**Figure 6.** A working model to explain the evolution of the amylase locus. The schematic diagram illustrates how the amylase gene locus could have independently evolved in different lineages (Lineage A and B) of the mammalian phylogeny. Our results are consistent with a model where the enzymatic expression in saliva is gained through the concerted effect of gene duplications and regulatory architecture. Our findings suggest that at least one gene duplication is required for salivary-gland-specific expression. Our suggested model posits that along with the strength of the regulatory architecture and the dietary context, additional gene duplications drive the evolution of amylase expression in saliva. Examples representative of Lineage A are *muroidea* and *cercopithecinae*. Examples representative of Lineage B are dogs, pigs, and humans.
DOI: https://doi.org/10.7554/eLife.44628.011

Based on our findings, we propose here that at least one gene duplication is required for amylase to gain expression in salivary glands. We argue that this is a form of neofunctionalization (*Ohno S, 1970*) where an otherwise intact gene copy acquires mutations in its regulatory architecture, thus leading to expression in a new tissue, in this case the salivary gland system.

We hypothesize that one or more gene duplication events of this copy are needed to gain a level of enzymatic activity in saliva optimally suited to accommodate starch predigestion in the mouth environment. As illustrated in *Figure 6*, we further propose that there are different ways to achieve a given level of salivary amylase activity, depending on the strength of the regulatory element associated with the salivary gland specific gene copy. For instance, *muroidea* and *cercopithecinae* possess only a single salivary-gland-specific gene copy but arrive at high levels of amylase activity in their saliva. In this case, we surmise that a strong regulatory element is associated with that single copy. In addition to that copy, *muroidea* possess multiple other amylase gene copies that are not expressed in salivary gland tissue. It can be assumed that these gene copies are expressed in pancreatic tissue and may increase enzymatic activity there in the more distal parts of the digestive tract. Dogs and pigs have numbers of amylase gene copies comparable to humans and some *muroidea* but show much less amylase activity in their saliva than these species. Given the data, we are not able to say which of these copies are expressed in pancreas and which in salivary glands. Regardless, those copies that might be expressed in salivary glands must be associated with weak regulatory architectures.

In humans, the number of amylase gene copy numbers was shown to correlate with levels of amylase expression in saliva (*Bank et al., 1992*; *Perry et al., 2007*). Our data show that across all mammalian species such a simple correlation does not hold true. However, within species possessing multiple and variable copy numbers there might be a correlation of copy number with salivary expression. In this regard, we could show a correlation of amylase gene copy number and salivary enzyme activity for dog ($R^2 = 0.45$) and pig ($R^2 = 0.69$). A comparative study investigating expression of individual amylase gene copies in pancreas versus salivary gland system will be the logical next step to delineate the mechanisms through which gland-specificity of amylase has evolved in mammals. It is of particular interest to simultaneously elucidate the role of lineage-specific retrotransposons or other regulatory elements in modulating tissue-specific expression of amylase.

From a broader ecological perspective, we showed that amylase gene copy numbers generally correlate with a broad-range diet and with high versus low-starch consumption. However, salivary amylase activity is only correlated with broad-range diet but not with the amount of starch. As such, a simple explanation solely based on the digestive function of amylase cannot fully explain why some mammals, including humans, express so much amylase in their saliva. As a matter of fact, most mammalian species do not keep food long enough in their mouth for salivary amylase to substantially participate in starch digestion. We argue here that in most mammalian species the role of amylase in saliva may be to liberate oligomeric sugar molecules from polymeric starch chains that can then be perceived by sweet-taste receptors in the oral cavity. Indeed, studies found links between salivary amylase and taste perception (*Mandel et al., 2010*; *Sclafani et al., 1987*). Being able to perceive otherwise tasteless starch in their diet might confer an adaptive advantage to species consuming broad-range diets by enabling them to detect high caloric (i.e. starch-containing), food components. Lastly, putatively adaptive benefits of amylase expression in saliva depend on the ecological and behavioral context for any given species. An exceptional example are the cheek-pouched *cercopithecinae* where putative fitness advantage of salivary amylase expression goes beyond taste perception. In these species, which conduct almost half of their starch digestion in the oral cavity (*Janiak, 2016*), salivary amylase may have evolved to substantially participate in the overall digestion of dietary starch, a role executed by pancreatic amylase in most other species.

In addition to gustatory and digestive functions, salivary amylase may also be involved in the regulation of metabolic glucose homeostasis (*Peyrot des Gachons and Breslin, 2016*) as well as associated with bacterial composition in the oral cavity (*Davenport, 2017*; *Scannapieco et al., 1989*) or in the gut (*Poole et al., 2019*). In that regard it will be interesting to find out whether different amylase gene copies encode proteins of slightly different functional activity due to differences in DNA sequence, differential RNA splicing, and post-translational modifications, including glycosylation. Overall our present study highlights the potential role of amylase enzymatic activity in saliva in shaping food preference and niche partitioning among omnivorous starch-consuming mammals, possibly in coevolution with the oral microbiome.

## Materials and methods

### Sampling

We chose our panel of mammalian species based on their phylogeny, diet preference (broad vs. specialized), domestication status, and commensal relationship with humans. Overall, we compiled 204 DNA samples from 46 different species and 127 saliva samples from 22 different species. Detailed information about the samples used in this study and their sources can be found in *Supplementary file 1*. Briefly, DNA from various animals was collected using buccal swabs (PurFlock, Puritan Medical Products), saliva samples, or museum specimens (dried blood and tissues, Museum of Southwestern Biology, Division of Genomics Resources). Saliva samples were collected by suction using commercially available devices containing absorbent sponges in a syringe-like receptacle (Super-SAL and Micro-SAL, Oasis Diagnostics, Vancouver, WA) unless otherwise specified. DNA was extracted from swabs using a commercially available kit (ChargeSwitch gDNA Buccal Cell Kit, Invitrogen). For saliva samples, we used a commercially available extraction kit (BioWorld, Dublin, OH). Detailed information about sampling strategies employed for each species can be found in Collection of saliva samples section below.

### Copy number variation analysis

Digital droplet PCR (ddPCR) was used to experimentally determine amylase gene copy numbers. If reference genomes were available for a given species, primers were designed specifically for use in these species. For species where reference genomes were unavailable, amylase coding sequences were chosen for primer design that were confirmed to be conserved in the two most closely related species. Further details about primer design and strategy are described in Primer design for digital PCR section below. The primer sets used for each species are listed in *Supplementary file 2*.

## Phylogenetic analysis

Translated amino acid sequences of the amylase gene copies were downloaded from NCBI reference genomes. Sequences were aligned and a phylogenetic output was generated using a custom *Python* code as described previously (https://github.com/duoduoo/VCFtoTree) (*Xu et al., 2017*; *Pajic et al., 2016*). We constructed a phylogenetic tree from the protein sequences by Randomized Axelerated Maximum Likelihood (RAxML) (*Stamatakis, 2014*) using the LG substitution model (*Le and Gascuel, 2008*), bootstrapping it with 1000 replicates for branch support. Visualization was performed using the FigTree software (*Rambaut, 2012*).

## Retrotransposon analysis

Previous work utilized lineage-specific retrotransposons to estimate the timing of amylase gene duplications and to distinguish between salivary and pancreatic amylase genes in humans (*Samuelson et al., 1990*). Using this approach, the salivary *AMY1* gene could be traced back to a great ape ancestor (*Samuelson et al., 1990*). Later studies used the sequence of the great-ape specific retrotransposon to label *AMY1* gene copies in humans by fiber-FISH (*Perry et al., 2007*). Building upon these findings, we searched 5 kb upstream and downstream of the amylase copies in mouse, rat, pig, and dog reference genomes for the existence of lineage-specific retrotransposition markers. Specifically, we searched for relatively recent L1 elements having sW scores of more than 1000 and being located at nearly identical distances to the 5' or 3' ends of amylase gene copies. Using this approach, we detected a relatively small number (less than 10) of distinct L1 retrotransposons in each species (*Figure 2B*). To ensure that these retrotransposons were duplicated specifically in the amylase locus, we conducted a BLAST analysis to search for the existence of these same retrotransposons outside of the amylase locus. We found that the retrotransposons within the amylase locus are highly similar to each other (>90%) and we found no similarly close matches for these retrotransposons elsewhere in the reference genomes of these species (*Supplementary file 3*). Next, we verified that these retrotransposons were indeed lineage-specific by showing that there were no close matches in reference genomes of other species. The most parsimonious explanation for our observations is that these L1 elements inserted into the proximity of an amylase gene copy and then duplicated along with additional copies of that gene, thereby suggesting lineage-specificity of duplication events.

## Measurement of amylase enzymatic activity

We used two different methods to measure enzyme activity of amylase in saliva. First, we conducted a direct estimate of enzyme activity using a traditional starch lysis agar plate assay following a previously described protocol (*Kilian and Nyvad, 1990*). In brief, holes were punched in a starch-containing agar and filled with saliva. After 24 hr incubation at 37°C, the undigested starch remaining in the agar was stained with iodine and the diameters of the lysed clear rings were measured. Enzymatic activity was extrapolated from serial standard dilutions of purified α-amylase from human saliva (Sigma) measured in the same assay. In parallel, we measured the samples using a high-sensitivity (detection limit 2 mU/ml) colorimetric in-solution assay (EnzCheck *Ultra* Amylase Assay Kit, Invitrogen) following the manufacturer's protocol with the same human α-amylase as the standard. Concentrations of total protein in saliva were determined by the bicinchoninic acid (BCA) assay (micro-BCA, BioRad) using bovine serum albumin as the standard. Optical density measurements were performed using a Nanodrop 2000 spectrophotometer (Thermo Fisher). Amylase activities were calculated as units of enzymatic activity normalized per mg of total salivary protein.

## Data analyses

All input data used for creating the main figures are provided in *Supplementary file 1*. Information about the dietary preferences of individual species was acquired from extensive literature research presented in Categorization of starch consumption section below. All figures were produced using the R statistical package (https://www.r-project.org/). For calculating the independent phylogenetic contrasts shown in *Figure 4C*, we used the approach outlined by *Felsenstein (1985)*. For this analysis, we used the subset of species available through the Hg19 100way conservation alignment (http://genomewiki.ucsc.edu/index.php/Hg19_100way_conservation_alignment). Using the phylogenetic distance provided in this dataset, we first normalized the differences in amylase gene copy

number and salivary enzyme activity between any two species by the square roots of the phylogenetic distance between them. Using these normalized values and applying the non-parametric Kolmogorov–Smirnov test, we tested the null hypothesis that the phylogenetic contrasts between species consuming a specialized diet is not different from the phylogenetic contrasts between species consuming the different types of diet. Among the species consuming a broad-range diet, we further tested that the phylogenetic contrasts between high and moderate starch consuming species are not different from those between species consuming moderate levels of starch.

## Collection of saliva samples

Saliva samples and buccal swabs from deer mice (*Peromyscus spp.*) were provided by Danielle Garneau (SUNY Plattsburgh). Mice were trapped in the wild by *Sherman* live traps (*Garneau et al., 2012*). After restraint by scruffing behind the neck, a glass capillary tube was introduced to the animal's mouth and was moved about the lower lip and cheeks to collect saliva. The tube was introduced at an angle such that gravity would help draw down the sample into the tube. The capillary tube was placed in an Eppendorf tube and a pipet pump was used to force air to drive the rest of the sample from the capillary tube into the Eppendorf tube for storage at −20°C and shipment on dry ice.

Saliva from house mice (laboratory strain C57BL10/SNJ) was kindly provided by Jill Kramer (University at Buffalo) using a collection procedure as previously described (*Kiripolsky et al., 2017*).

Saliva from woodrats was kindly provided by Michelle Skopec (Weber State University). To collect saliva, woodrats were scruffed and Micro-Sal collection (Oasis Diagnostics, Vancouver, WA) devices were placed in their mouths. The woodrats were allowed to chew on the absorbent sponge part of the device and, then, their tongues and cheeks were swabbed to retrieve residual saliva. Collection devices were centrifuged and saliva samples were stored at −20°C before shipment on dry ice.

Saliva from Long Evans hooded rats was kindly provided by Ann-Marie Torregrossa (University at Buffalo). As described previously (*Martin et al., 2018*; *Torregrossa et al., 2014*), rats were conditioned to salivate when a pipette was inserted into the mouth and saliva was collected. Approximately 50 µl saliva was retrieved by suction from below and around the tongue where it pools naturally.

Saliva from dogs, cows, sheep, goats, horses, pigs, and giant African pouched rats was provided by Erin Daugherity and Luce E. Guanzini (Cornell University). Animals were not allowed to eat or drink prior to the collection to ensure the oral cavity was free of food and other debris. Saliva from giant African pouched rats was collected opportunistically while animals were anesthetized for an unrelated clinical procedure. The collection was performed using a commercially available device (Micro-Sal, Oasis Diagnostics). Large animals were gently restrained and a larger collection device (Super-Sal, Oasis Diagnostics) was placed under the tongue for up to three minutes, or until fully soaked. Devices were stored at −20°C before shipping on dry ice.

Saliva from female wild boars and castrated domestic pigs was provided by Anja Globig (Friedrich-Loeffler-Institut, Insel Riems - Greifswald, Germany). For the collection of saliva a commercial collection device, consisting of an absorbent cotton swab in a tube, was used (Salivette, Sarstedt, Nümbrecht, Germany). The swab was inserted in the animal's mouth and fixated with a forceps until it was drenched with saliva. After placing the swab back in the tube, saliva was extracted by centrifugation. Samples were lyophilized before international shipping.

Saliva from wolves was kindly provided by Karen Davis (Wolf Park, Battle Town, IN). The wolves housed in this facility are well socialized, which allowed saliva collection by inserting Super-Sal (Oasis Diagnostics) devices into the mouths of adult wolves willing to participate. Swabs were kept in the animals' mouths as long as they would tolerate it or until fully soaked. Samples from juvenile wolves could be collected while they were resting by inserting the swabs into their mouths. Samples were stored at −20°C before shipment on dry ice.

Saliva from dogs was kindly provided by Barbara McCabe (Buffalo, NY). Samples were obtained from diverse breeds of dogs including Boxers, Pitbulls, Golden Retrievers, and Labradors, along with several mixed breeds (see *Supplementary file 1* for details). Super-Sal devices (Oasis Diagnostics) were placed in the mouth of dogs for 1–5 min, or until swab was damp. The swabs were stored at −20°C until transfer to our laboratory.

Saliva from Ring-tailed Lemur samples was kindly provided by Erin Ehmke (Duke Lemur Center). Samples were collected using commercially available absorbent strips (SalivaBio Children's Swabs,

Salimetrics, Carlsbad, CA). Saliva-soaked swabs were immediately centrifuged and the collected saliva was frozen at −80°C and shipped on dry ice.

Saliva from humans was collected by passive drooling following the protocol approved by the University at Buffalo Human Subjects IRB board (study # 030–505616). Informed consent was obtained from all human participants. Saliva from chimpanzees and gorillas was collected in a noninvasive manner following the protocol approved by the University at Buffalo IACUC committee (IACUC ID# AR201800024). Chimpanzees were trained by the caretaker to voluntarily expectorate into a plastic cup. Gorilla (Western lowland gorilla) saliva was collected by the animal caretakers with a soft disposable plastic Pasteur pipette (VWR, Radnor, PA) from individuals who were previously trained to open their mouth upon request. Saliva from Rhesus macaques was provided by the Southwest National Primate Research Center, San Antonio, TX, and by the Yerkes National Primate Research Center, Atlanta, GA. All samples were immediately transferred into a polypropylene tube chilled on ice. Aliquots were stored at −80°C and shipped on dry ice.

## Categorization of starch consumption

All input data used for creating the main figures are provided in *Supplementary file 1*. Information about the dietary preferences of individual species was acquired from the Michigan Animal Diversity Web (https://animaldiversity.org/), unless other studies were cited. With regard to starch consumption, the literature was limited. Thus, we undertook the following steps to construct a categorization of starch consumption among the species that we used in our analysis presented in *Figure 4A and B*.

Based on the information available on Michigan Animal Diversity Web, we first identified animals with specialized diets (*carnivores*: cat, polar bear, cougar, and wolf; *herbivores*: sheep, bison, snow sheep, cow, goat, horse, bighorn sheep, ibex, yak, wild goat, zebra, sheep, and donkey). We assumed that starch makes up a negligible percentage of these animals' diet.

For the animals with broad-range diets, which presumably have considerable starch content in their diet, we conducted a wider literature research. Most information about starch consumption is available for present-day human populations (*Bright-See and Jazmaji, 1991*), and it has been suggested that humans consume a higher percentage of starch in their diet than great apes (*Perry et al., 2007*; *Zohary et al., 2012*). Indeed, chimpanzees and bonobos primarily consume ripe fruits (poor in starch), and in the scarcity of ripe fruits, they prefer piths (also poor in starch) (*Hohmann, 2009*; *Wrangham et al., 1998*). Orangutans and gorillas primarily consume ripe fruit and leaves (*Hohmann, 2009*). However, it was suggested that they also consume seeds and cambium, an observation that led Janiak (*Janiak, 2016*) to argue that gorillas and orangutans have a relatively higher starch content in their diets than chimpanzees and bonobos.

Old World monkeys show remarkable diversity in their diets. Specifically, cercopithecines (represented in our dataset by baboons, rhesus macaques, vervet monkeys, green monkeys, and Allen's swamp monkeys) consume considerable amounts of unripened fruit (higher starch content [*Lambert, 1998*]), especially when no ripe fruit is available (*Mau et al., 2010*; *Wrangham et al., 1998*). In contrast, most *colobinae* (represented by the colobus monkeys in our dataset) are primarily leaf-eating and, thus, likely have little starch in their diet (*Oates, 1994*).

New World monkeys (represented by capuchin monkeys, owl monkeys, and marmosets in our dataset) also show diversity in their likely starch consumption. Capuchin and owl monkeys primarily consume fruits, even though they supplement their diets with flower foraging and insects (*Lambert, 2017*; *Kinzey, 1997*), which likely indicates starch consumption similar to chimpanzee and bonobos. Marmosets differ in their dietary habits as their primary food intake comprises gum and other exudates (which are not starch sources) from various trees and vines, and scarcely involve fruits (*Soini, 1982*). Lemurs (represented by ring-tailed lemur in our dataset) have been reported to primarily consume Tamarind fruit, which is rich in non-starch polysaccharides (*Gould et al., 2003*). Overall, primates depend on a wide variety of food sources, including starch-based foods. However, humans are the only primate species consuming a diet unusually high in starch content.

As for rodents, we first considered species which are primarily human-commensal (represented by house mice as well as by brown and black rats in our dataset). These species have considerable variation in their diets depending on the ecological context they are living in. For example in the wild, house mice have been reported to eat primarily insects and seeds, the latter containing significant amounts of starch (*Badan, 1986*; *Roux et al., 2002*). However, house mice normally live in

human-influenced habitats and consume agricultural grains and other starch-rich human-produced food and human food leftovers (*Clark, 1982*; *Gardner-Santana et al., 2009*; *Hulme-Beaman et al., 2016*; *Pocock et al., 2004*; *Schein and Orgain, 1953*; *Singleton et al., 2003*). Other rodents with less human-commensal interactions (represented by birch mouse, deer mouse, pouched rat, and woodrat in our study) also consume diverse diets, including insects, seeds, grains, and flowers (*Ajayi, 1977*; *Baker, 1991*; *Everett et al., 1978*; *Juskaitis, 2000*). However, their access to starch-rich grains and seeds is seasonal, and these foods are not necessarily their primary caloric source.

The other mammals with broad-range diet in our dataset were dogs, pigs, boars, and bears. Brown and black bears eat a wide range of foods including leaves, fruits, grains, as well as meat from hunting or scavenging (*Bojarska and Selva, 2012*; *Graber and White, 1983*; *Torgersen et al., 2001*). However, the same studies documented that starch-containing foods, such as grains, make up only a small portion of the bears' diet. Boars also have a diverse preference in their diet, including mushrooms, roots, fruits, and insects. However, unlike bears, boar diets include substantial amounts of roots and tubers (*Baubet et al., 2004*; *Massei and Genov, 2004*) and, if available, human agricultural crops (*Herrero et al., 2006*). A close relative of the boar, the domesticated pig thrives primarily on human-produced starch-rich food sources, such as corn or potatoes. Overall, pigs and boars have higher starch content in their diets than bears. In fact, their diet can be comparable to early human diets (*Hatley and Kappelman, 1980*; *Miller and Ullrey, 1987*). Another mammal consuming a broad-range diet that was included in our study was the dog, which has been discussed within the context of recent adaptation to human-derived starch-rich diets (*Arendt et al., 2016*; *Axelsson et al., 2013*). Based on this literature, we presume that dogs, pigs, and boars have higher starch content in their diet, while starch makes up a smaller portion of the diet of the bears.

Overall, all the mammals consuming a broad-range diet mostly have considerable levels of starch in their diet. However, our literature search indicates that humans, pigs, boars, dogs, mice, and rats (both brown and black) stand out in that their diet predominantly depends on starch-rich foods (grains, roots, and tubers). Thus, we grouped them under the 'higher starch' consuming category whereas we grouped the other species under the 'lower starch' consuming category (*Figure 4A* and *B*).

## Primer design for digital PCR

For digital droplet PCR experiments, we used two primer/probe sets. One targeted the amylase copies and the other targeted a conserved 'reference' sequence, which was found to be a single haploid copy in mammals with known reference genomes (SRSF7 gene). For the reference sequence, we used a primer/probe set that targets one of the exons of the *SRSF7* gene. The sequence is highly conserved across species and unique (i.e. a single haploid copy) in all mammalian reference genomes we investigated. We have checked that the sequence is 100% conserved in species that we considered and for which reference genomes were available from the UCSC genome portal. To capture as many amylase gene copies as possible, we carefully designed primers and probes for each species where a reference genome was available to match (100% as assessed by BLAST alignment) all of the reference amylase copies. Primer and probe sequences are listed in *Supplementary file 2*. It is possible that we underestimated or missed some of the amylase copies that are not represented in the reference genomes. However, digital PCR is robust to 1–2 mismatches and in most species, ddPCR results were highly concordant with copy number estimations based on BLASTx and BLASTp analysis (*Figure 1—figure supplement 1*, *Supplementary file 2*). Therefore, we surmise that the main trends we observed in this study are reliable.

To decide which primer/probe sets to use for species where no reference genome was available, we designed primer/probe sets that work in the phylogenetically most closely related species for which reference genomes were available. For example, for zebra, we used a primer/probe set designed for the horse reference genome. To ensure that this primer/probe set was appropriate, we first made sure that it also worked in the donkey reference genome. As such, we surmised that our approach should work unless there is rapid, species-specific sequence divergence in zebra as compared to horse and donkey. An analogous approach was used for all the other species for which reference genomes were not available (*Figure 1—figure supplement 1*, *Supplementary file 2*, for substitute species genomes chosen). Of course, this approach might be prone to undercalling the number of amylase gene copies in species where reference genomes are not available. Although we

are confident about these estimates, none of the major conclusions of this study depends on data from such species.

## Acknowledgements

We recognize and thank all individuals and institutions who provided us with DNA and saliva samples: Joseph Cook and Mariel Campbell at the Museum of Southwestern Biology, Division of Genomics Resources, University of New Mexico; the Coriell Institute for Medical Research, Camden, New Jersey; Karen Davis at Wolf Park, Battle Ground, Indiana; Barbara McCabe, University at Buffalo, New York; Luce E Guanzini at the Center for Animal Resources and Education, Cornell University; Klaus Depner at the Friedrich-Loeffler-Institut, Greifswald, Germany; Ann-Marie Torregrossa, Department of Psychology, University at Buffalo; Michele Skopec at the Department of Zoology, Weber State University, Ogden, Utah; Jill Kramer at the Department of Oral Biology, University at Buffalo; Kurt Volle, Alicia Dubrava and fellow gorilla caretakers of the Buffalo Zoo, the Southwest National Primate Research Center (funded by NIH - ORIP/OD P51 OD011133), the Yerkes National Primate Research Center (funded by NIH - ORIP/OD P51 OD011132), and the Duke Lemur Center (funded by NSF - LSCBR #1050035). We are grateful to Ozgur Taskent, Jessica Poulin, Trevor Krabbenhoft, and Derek Taylor for proofreading the manuscript and discussions of the data. This study was funded by The National Science Foundation NSF grant No. 1714867 (OG), National Institute of Dental and Craniofacial Research (NIDCR) grants R01 DE019807 and R21 DE025826 (SR), and National Cancer Institute (NCI) grant U01 CA221244 (SR).

## Additional information

### Funding

| Funder | Grant reference number | Author |
|---|---|---|
| National Institute of Dental and Craniofacial Research | R01 DE019807 | Stefan Ruhl |
| National Institute of Dental and Craniofacial Research | R21 DE025826 | Stefan Ruhl |
| National Cancer Institute | U01 CA221244 | Stefan Ruhl |
| National Science Foundation | 1714867 | Omer Gokcumen |

The funders had no role in study design, data collection and interpretation, or the decision to submit the work for publication.

### Author contributions

Petar Pajic, Conceptualization, Data curation, Formal analysis, Validation, Investigation, Visualization, Methodology, Writing—original draft, Writing—review and editing; Pavlos Pavlidis, Formal analysis, Methodology, Writing—review and editing; Kirsten Dean, Validation, Investigation; Lubov Neznanova, Validation, Investigation, Methodology; Rose-Anne Romano, Resources, Data curation, Methodology; Danielle Garneau, Resources, Methodology, Writing—review and editing; Erin Daugherity, Resources, Validation, Methodology; Anja Globig, Resources, Validation, Methodology, Writing—review and editing; Stefan Ruhl, Conceptualization, Supervision, Funding acquisition, Visualization, Methodology, Writing—original draft, Project administration, Writing—review and editing; Omer Gokcumen, Conceptualization, Resources, Formal analysis, Supervision, Funding acquisition, Visualization, Methodology, Writing—original draft, Writing—review and editing

### Author ORCIDs

Stefan Ruhl http://orcid.org/0000-0003-3888-4908
Omer Gokcumen http://orcid.org/0000-0003-4371-679X

## Ethics

Human subjects: Saliva from humans was collected by passive drooling following the protocol approved by the University at Buffalo Human Subjects IRB board (study # 030-505616). Informed consent was obtained from all human participants.

Animal experimentation: Saliva from chimpanzees and gorillas was collected in a noninvasive manner following the protocol approved by the University at Buffalo IACUC committee (IACUC ID# AR201800024). The samples from other animals were collected in collaboration with colleagues at museums, zoos, and other research institutions. The samples from all the live animals specifically for this study were collected using minimally invasive methods, and involved buccal swabs and saliva collections with specialized kits or from the drool. Detailed descriptions of sample sources and collection methods can be found in the main text.

## Decision letter and Author response

Decision letter https://doi.org/10.7554/eLife.44628.017
Author response https://doi.org/10.7554/eLife.44628.018

# Additional files

## Supplementary files

• Supplementary file 1. Primary datasets used in our study including amylase enzymatic activity in saliva and gene copy numbers across mammalian species (tab 1: data used for *Figures 3* and *5*), as well as dietary preferences across species (tab 2: data used for *Figure 4*), and amylase copy numbers found in available reference genomes (tab 3: data used for *Figure 1—figure supplement 1*).
DOI: https://doi.org/10.7554/eLife.44628.012

• Supplementary file 2. Details of the primer and probe sets that we used in this study.
DOI: https://doi.org/10.7554/eLife.44628.013

• Supplementary file 3. Results of our analysis of retrotransposons associated with the amylase gene copies depicted in *Figure 2B*. Figure supplements
DOI: https://doi.org/10.7554/eLife.44628.014

• Transparent reporting form
DOI: https://doi.org/10.7554/eLife.44628.015

## Data availability

All data generated (copy numbers and salivary activity information) can be found in the supplementary files and Methods.

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
