## [Decision Letter]

[Editors’ note: a previous version of this study was rejected after peer review, but the authors submitted for reconsideration. The first decision letter after peer review is shown below.]

Thank you for submitting your work entitled "*Amylase* copy number analysis in several mammalian lineages reveals convergent adaptive bursts shaped by diet" for consideration by *eLife*. Your article has been reviewed by four peer reviewers, including George H Perry as the Reviewing Editor and Reviewer #1, and the evaluation has been overseen by a Senior Editor.

Our decision has been reached after consultation between the reviewers. Based on these discussions and the individual reviews below, we regret to inform you that your work will not be considered further for publication in *eLife*.

There is general agreement among the reviewers (including me as the reviewing editor) that multiple results within your dataset would be of fundamental interest to the *eLife* genomics and evolutionary biology readership communities. However, we collectively identified (and shared a consensus opinion about, following consultation) multiple substantial issues with the current version of the analyses presented and the manuscript that preclude our ability to consider your submission for publication at this time. In particular, the phylogenetic and evolutionary analyses presented did not account for gene conversion and phylogenetic non-independence and important data that should be available for inclusion are missing, interpretations and conclusions about the relationship between copy number and salivary amylase expression are (perhaps unnecessarily) over-extended, and there are questions about the accuracy of the cross-species ddPCR approach that are not alleviated by the supplementary QC figure. These and other substantial concerns are detailed in the individual reviews below.

In addition, I want to draw your attention to a different, but also important, type of major limitation in the present manuscript, specifically the (much too limited) amount of methodological and analytical detail provided. This issue frustrated peer experts in these methods and in this area of research and precluded our thorough review of many aspects of the paper. This would be an even bigger problem for a general readership.

Given our interest in the potential of this dataset, I do not want to completely slam the door on consideration of your manuscript in the future at *eLife*, if you are able to substantially re-work the analyses and manuscript to address the major concerns raised through this process and find that your primary conclusions are supported more robustly. However, we feel that this revision would require an extensive amount of work, and moreover as I mentioned above there are multiple components of even the present version of the paper that we could not yet assess fully. Thus any future submission would be considered an entirely new manuscript, with high expectations at the editorial review stage and no guarantee of full review. Regardless of how you choose to proceed, we hope that the detailed comments provided below are helpful for your next round of revision of this interesting dataset, and we look forward to seeing the ultimate outcome!

Reviewer #1:

This manuscript from Pajic et al. presents a broad survey of amylase gene (*AMY*; the protein products of *AMY* genes help digest dietary starch) copy number variation among mammals (153 individuals from a total of 44 species), alongside salivary amylase protein expression level data for a subset of those taxa (118 individuals from a total of 20 species). The paper extends previous datasets available for humans and dogs to identify the intriguing result of widely recurrent *AMY* copy number gains in the genomes of species with relatively higher levels of dietary starch compared to related taxa (e.g. pigs, mice and rats, capuchins, cercopithecine monkeys, humans), although there are limitations associated with the absence of equivalent diet data for all of the species tested. Still, this broader pattern will likely be robust to necessary revisions to the evolutionary analysis and other aspects of the manuscript, although some other technical issues could undermine the underlying data depending on answers to some of the below questions about the experimental design.

Essential revisions:

1) The accuracy of the digital droplet PCR method to estimate amylase copy number across this broad range of species is critical. In the brief Materials and methods, it is stated "For primer design we targeted amylase exonic sequences that are conserved among copies and between species." But 11 different primer sets were used, and of course even within species groups (and among gene family copies within species, e.g. *AMY1* vs. *AMY2* in apes) sharing the same primer set varying levels of sequence divergence at these primer sites is expected. How does this impact the results? The methods for confirming the accuracy of this approach (other than reference to Supplementary Figure 4) are absent, and even for the analysis of Supplementary Figure 4 I suspect that the included species might be biased towards those from which the primer sequences for each group were designed in the first place, keeping this from being a true assessment.

2) The premise of the phylogenetic analysis of amylase coding region sequences to conclude that amylase duplications occurred independently within each lineage with duplications rather than being an ancestral trait (versus gene loss in some species instead) does not consider either gene conversion or ultra-high rates of NAHR, either of which (or both in combination) could obliterate any long-term phylogenetic signals in these data. I'm not sure that this is resolvable. While I agree that independent duplication events are the most likely scenario, and this is something that could be discussed as such, I don't think the authors' analyses or interpretations should be reliant on this demonstration. That is, the pattern is still evolutionarily interesting even if it is functional constraint to maintain higher copy numbers in lineages with higher levels of dietary starch, with losses in other lineages.

That said, I thought that the mouse vs. human retrotransposon result was convincing, and the rat versus mouse results may be as well, although this result needs more description and explanation in the Results text. The data/logic for the dog and pig/boar results are not provided (and again the text here seems to suggest that dogs and wolves diverged only 5000 years ago, which is incorrect), which needs to be addressed. This all can be presented as part of the 'We believe the most likely explanation for these observations are repeated, independent duplication events in each lineage… however, we cannot exclude… regardless, of evolutionary interest" text in the revised approach to this discussion.

3) In general (as mentioned directly above, but this is actually a wide problem and a big issue) there is insufficient detail presented on the experimental results and analyses in the paper. While this issue can readily be addressed in revision, it means that reviewers may not be able to identify some problems from this version of the manuscript.

4) The finding of salivary amylase expression in some dog breeds is definitely interesting. However, I felt that the conclusions of *AMY* duplication 'bursts' necessarily being linked to the production of salivary amylase was too strong based on the data presented, and again seemingly not necessarily for this to be an interesting and valuable paper.

5) Why are non-human primates analyzed separately from the non-primate mammal data?

6) In the evolutionary analyses, some sort of approach that takes phylogenetic relationships into account should be used. Also, I understand the limitations with differentially available dietary starch intake data among species… however, could a subset of the dataset be formally analyzed in this framework, i.e. considering the species for which roughly equivalent dietary data are available?

7) Details were insufficient for me to evaluate the *AMY* copy number simulation results, although it is difficult for me to imagine the parameters of the simulation exercise being able to accurately model mutation rates (and especially given that rates of NAHR once duplication has occurred are expected to be strikingly higher than the rate of initial duplication of a locus).

Reviewer #2:

In this paper, employing digital PCR experiments to assess amylase copy number across a diverse range of mammalian lineages, Pajic et al. identify higher copy number within certain starch-consuming lineages. Additionally, they perform assays to measure salivary amylase activity and coalescent simulations to model neutral copy number variant evolution.

This work is performed in the context of known amylase gene family expansions in human, rodents, and dog, so the novelty here is the comprehensive nature with which sampling and analysis took place. The data presented here is confirmatory of the previously known amylase gene family expansions without addressing some relevant previous work, notably positive selection identified to be acting on amylase loci in mouse and dog (Staubach et al., 2012, Reiter et al., 2016).

The possibility of gene conversion and the effect this could have on the data and inferences presented here is not discussed. There is evidence of gene conversion within this locus in human (Groot et al., 1990) and only the scenario of numerous gene duplication events and one gene loss event is presented in the section 'Evolution of amylase in primates'. If gene conversion is taking place, then duplicates may appear to be younger and more lineage-specific. Alternative scenarios are not discussed as other explanations of the data observed. Fewer and older duplication events could have occurred, followed by more than one gene loss event and this history is masked by lineage-specific similarity as a consequence of gene conversion. Additionally, ancestral duplication polymorphisms, incomplete lineage sorting and gene conversion is a possibility that is not adequately dismissed with the data and arguments presented.

The maximum likelihood phylogenetic tree constructed using amylase protein sequences translated from reference genomes (Figure 2A) shows species-specific clustering of sequences. The phylogenetic tree would be more informative if it also including sequences from species where increases in copy number have not occurred and was rooted with an outgroup, which are both readily available.

The authors state that "amylase gene copy numbers in species correlate well with observable enzymatic activity in saliva (Figure 3C)" and additionally state that "the copy annotated as mouse *AMY1* (Figure 2) is expressed in salivary glands, and is likely responsible for salivary expression of amylase in mice, while the other amylase duplicates have a negligible expression in salivary gland tissue (Figure 2—figure supplement 2)." However, they do not adequately reconcile these two statements. If only one copy in mouse is meaningfully expressed in saliva, what does that mean for salivary expression correlation with copy number? All but one copy have no impact on salivary activity so, for mouse at least, a relationship between salivary expression level and number of copies is not direct. Perhaps salivary expression level is correlated with a starch-rich diet and a starch-rich diet is also correlated with pancreatic expression. Increased pancreatic expression could be facilitated through gene duplication and increased salivary expression through enhancers? Discussing these alternative means of achieving increased expression in different tissues given the data observed would be welcome.

Reviewer #3:

In "Amylase copy number analysis in several mammalian lineages reveals convergent adaptive bursts shaped by diet" Pajic et al. examine amylase copy number across the mammalian phylogeny. The authors observe that amylase copy number appears to be higher in species that have high-start diets than those that do not, and that this correlates with levels of salivary amylase. Although these are some compelling observations, the authors claims of "adaptive bursts" and convergence in the title are both unjustified in my opinion.

Regarding "adaptive bursts": this is largely supported by the authors evidence of independent origins of amylase copies. However, the protein sequence tree shown in Figure 2 could be effected by interlocus gene conversion, which will cause copies in the same species to appear more closely related. Thus we cannot be certain of the extent of the "burstiness" of the duplication events. The authors have also performed simulations of gene family evolution under various mutation rates, but I found this analysis rather simplistic (e.g. no difference in gain and loss rates allowed), qualitative (no model fits are formally assessed), and incomplete (no alternative models including natural selection are considered).

On a related note, in Figure 5, it is not clear to me how the authors placed the locations of duplication events across the phylogeny, and how confident we can be in these placements regardless of whether gene conversion is taken into account. Overall I would say that while the authors' observations are interesting and highly suggestive, there is no direct evidence for positive selection in these data, and thus the language about adaptation sprinkled throughout the manuscript should be removed or rephrased.

Second, the evidence for convergent evolution is limited. The strongest example of convergence that the authors give is in the third paragraph of the subsection “Recurrent amylase copy number gains in multiple mammalian lineages”, where they describe that in both humans and mice there are tandem arrays of amylase genes with a flanking transposable element, suggesting the possibility of repeat-induced expansion of this gene family in both species. However, in mouse these copies do not encode a salivary amylase according to Figure 2. Thus this example provides no support for convergence related to salivary digestion.

Third, the authors examine in Figures 3 and 4 evidence of the relationship between amylase copy number and salivary amylase activity. However as far as I can tell the authors have not assessed the strength of this relationship statistically in a manner that accounts for phylogenetic relatedness among data points. Moreover, to me the authors' results imply that the amylase copy numbers are cannot explain a sizable portion of the variation in amylase activity among mammals. This is demonstrated in Figure 4, where the authors show amylase copy number along a linear scale and salivary amylase activity in log scale. For amylase copy number the difference between the "specialized" and "broad range" diet classes spans about a factor of 2- or 3-fold. For salivary amylase, the difference is about 2-3 orders of magnitude. Assuming that copy number has at best a linear effect on protein dosage, this result strongly argues against copy number having a very important role in salivary amylase activity in mammals.

Reviewer #4:

The authors seek to understand whether the evolution of additional *AMY* genes occurred independently in different mammalian lineages, and whether the driving force behind *AMY* gene expansion was a starch-rich diet. To do this, they use a multi-pronged approach to investigate their research questions, including enzymatic assays, ddPCR to measure *AMY* gene copy numbers, and estimating copy numbers from reference genomes. The resulting dataset is extensive, with data on salivary amylase expression and *AMY* copy number variation in a large sample of mammals, significantly expanding our current knowledge about the evolution of salivary and pancreatic amylase, as well as the link between digestive enzyme evolution and diet more broadly. The manuscript is well-written and the figures effectively communicate the findings.

The *AMY* gene copy numbers are quite consistent between ddPCR and reference genome estimates, providing confidence in the authors' findings regarding copy number variation.

Interestingly, the authors find salivary amylase expression in mammals that were previously found not to express amylase in their saliva, including New World primates and dogs. Given that previous research did not find evidence for salivary amylase expression in these species (Perry et al., 2007; Axelsson et al., 2013), I think some skepticism is warranted. The authors state that dogs, for example, express "substantial amounts" of amylase in their saliva, however, based on the data in Supplementary file 1, amylase activity was quite low in most dogs compared to humans and rodents. Similarly, amylase activity in capuchins, pigs, and boars seemed quite low compared to catarrhines and rodents, making me wonder if some of the amylase "activity" may have been contamination or a basic level of activity that is always present. Was a negative control used? If so, it would be helpful to include the activity measures obtained for the negative control in Supplementary file 1. If a negative control was not used, the authors should repeat the assays with one.

In the Results, the authors state that salivary amylase activity varied across different dog breeds and refer the reader to Supplementary file 1. I was curious to see which dog breeds have higher salivary amylase activity, however, Supplementary file 1 does not include information on the different dog breeds included in the sample. I think this information should be included and the differences might be worthy of additional discussion. Do breeds associated with high-starch human populations have higher salivary amylase activity, for example?

[Editors’ note: what now follows is the decision letter after the authors submitted for further consideration.]

Thank you for submitting your article "Independent amylase gene copy number bursts correlate with dietary preferences in mammals" for consideration by *eLife*. Your article has been reviewed by four peer reviewers, including George H Perry as the Reviewing Editor and Reviewer #1, and the evaluation has been overseen by Diethard Tautz as the Senior Editor.

The reviewers have discussed the reviews with one another and the Reviewing Editor has drafted this decision to help you prepare a revised submission.

Summary:

The reviewers commend your efforts to revise your manuscript in response to our original reviews. The manuscript has been re-worked substantially, to positive effect. The reviewers felt that your primary conclusion that there have been bursts of amylase gene duplication in multiple lineages independently was now supported more strongly. Overall, the combination of your careful molecular genetics work on this complex locus alongside the protein expression measurements across a diversity of mammals is impressive, providing a substantial new advance into the evolutionary biology and evolutionary ecology of the amylase gene family, with broader molecular biology insights related to gene duplication and tissue expression pattern processes. We do collectively share some concerns about the manuscript that we ask you to address thoroughly.

Essential revisions:

1) Clarify your methods for grouping broad-range diet mammals into lower-starch and higher-starch categories. We sympathize with the challenges of obtaining equivalent food consumption data for all species, but greatly increased detail is necessary here. This analysis should be fully transparent and replicable.

2) The simulations as presented are uncompelling. If the authors choose to maintain this component of the manuscript, then the simulation approach and mechanics, process of parameter selection (using a range of values encompassing extremes of reasonable possibility, with clear justifications), and results all need to be clarified and detailed much more extensively. Is the diet information included in the model (i.e. asking how often neutral processes can explain the observed pattern whereby species with higher-starch diets are much more likely to have duplication bursts)?

3) Explicitly present a more complete/clarified model (or models) for amylase duplication and the gain of salivary amylase that could produce the observations in the paper. Some thoughts along these lines: If the model is simply that duplication must occur prior to the acquisition of salivary expression, then this is essentially a model of neofunctionalization (here, convergent in multiple mammal lineages for which starch is a substantial part of the diet). This model would posit that a single amylase copy cannot do the job of pancreatic and salivary amylase, perhaps due to changes to the amino acid sequence required for proper amylase activity (this is unlikely?), or instead that something about the regulatory structure of amylase prevents expression of a single copy in both the pancreas and in salivary glands. Also, the authors' results and interpretation now implies that just one new gene copy is required for the acquisition of sufficient salivary amylase activity, so the presence of additional copies in some species should be explicitly addressed as part of this discussion. We recommend creating a clearly defined section (e.g. 'Model for amylase gene locus evolution' or similar) in which this discussion can be concisely presented (note that a few relevant existing statements along these lines can be moved and modified from other sections), without needing to be conclusive. Opportunities for future investigation can be noted.

---

## [Author Response]

[Editors’ note: the author responses to the first round of peer review follow.]

There is general agreement among the reviewers (including me as the reviewing editor) that multiple results within your dataset would be of fundamental interest to the eLife genomics and evolutionary biology readership communities. However, we collectively identified (and shared a consensus opinion about, following consultation) multiple substantial issues with the current version of the analyses presented and the manuscript that preclude our ability to consider your submission for publication at this time. In particular, the phylogenetic and evolutionary analyses presented did not account for gene conversion and phylogenetic non-independence and important data that should be available for inclusion are missing, interpretations and conclusions about the relationship between copy number and salivary amylase expression are (perhaps unnecessarily) over-extended, and there are questions about the accuracy of the cross-species ddPCR approach that are not alleviated by the supplementary QC figure. These and other substantial concerns are detailed in the individual reviews below.In addition, I want to draw your attention to a different, but also important, type of major limitation in the present manuscript, specifically the (much too limited) amount of methodological and analytical detail provided. This issue frustrated peer experts in these methods and in this area of research and precluded our thorough review of many aspects of the paper. This would be an even bigger problem for a general readership.Given our interest in the potential of this dataset, I do not want to completely slam the door on consideration of your manuscript in the future at eLife, if you are able to substantially re-work the analyses and manuscript to address the major concerns raised through this process and find that your primary conclusions are supported more robustly. However, we feel that this revision would require an extensive amount of work, and moreover as I mentioned above there are multiple components of even the present version of the paper that we could not yet assess fully. Thus any future submission would be considered an entirely new manuscript, with high expectations at the editorial review stage and no guarantee of full review. Regardless of how you choose to proceed, we hope that the detailed comments provided below are helpful for your next round of revision of this interesting dataset, and we look forward to seeing the ultimate outcome!

We are grateful for your careful consideration of our paper and your thoughtful and thorough comments. We took them to heart and conducted several additional analyses to address each of these concerns. We also want to apologize for the lack of details in our originally submitted Materials and methods section. We now thoroughly revised and expanded our Materials and methods section. We also included a new Materials and methods section to include more specific details concerning our approach. Furthermore, to better reflect the revised content of this manuscript, we modified the main title to “Independent amylase gene copy number bursts correlate with dietary preferences in mammals”.

Reviewer #1:Essential revisions:1) The accuracy of the digital droplet PCR method to estimate amylase copy number across this broad range of species is critical. In the brief Materials and methods, it is stated "For primer design we targeted amylase exonic sequences that are conserved among copies and between species." But 11 different primer sets were used, and of course even within species groups (and among gene family copies within species, e.g. AMY1 vs. AMY2 in apes) sharing the same primer set varying levels of sequence divergence at these primer sites is expected. How does this impact the results? The methods for confirming the accuracy of this approach (other than reference to Supplementary Figure 4) are absent, and even for the analysis of Supplementary Figure 4 I suspect that the included species might be biased towards those from which the primer sequences for each group were designed in the first place, keeping this from being a true assessment.

The reviewer raises a salient point which we had actually considered when originally designing our primers. We apologize that in our original submission we did not thoroughly explain our strategy in primer and probe design. We now added more information to the Materials and methods section and provide a detailed description of the primer design and choices of probes for the various mammalian species investigated. We also expanded Supplementary file 2to now show the degree to which each given primer pair matches the targeted sequence in the different species. We further provide a new supplementary figure (Figure 1—figure supplement 1) to explain how we designed primers for species where no reference genomes were available. For such cases, we inferred the sequences from reference genomes of phylogenetically most closely related species. We now also completed direct genotyping of *AMY* gene copy numbers by ddPCR for those samples to show a direct measurement of copy number and salivary activity of amylase in all samples in Figure 3C. We now believe that our current dataset is robust enough to support our main conclusions.

2) The premise of the phylogenetic analysis of amylase coding region sequences to conclude that amylase duplications occurred independently within each lineage with duplications rather than being an ancestral trait (versus gene loss in some species instead) does not consider either gene conversion or ultra-high rates of NAHR, either of which (or both in combination) could obliterate any long-term phylogenetic signals in these data. I'm not sure that this is resolvable. While I agree that independent duplication events are the most likely scenario, and this is something that could be discussed as such, I don't think the authors' analyses or interpretations should be reliant on this demonstration. That is, the pattern is still evolutionarily interesting even if it is functional constraint to maintain higher copy numbers in lineages with higher levels of dietary starch, with losses in other lineages.That said, I thought that the mouse vs. human retrotransposon result was convincing, and the rat versus mouse results may be as well, although this result needs more description and explanation in the Results text. The data/logic for the dog and pig/boar results are not provided (and again the text here seems to suggest that dogs and wolves diverged only 5000 years ago, which is incorrect), which needs to be addressed. This all can be presented as part of the 'We believe the most likely explanation for these observations are repeated, independent duplication events in each lineage… however, we cannot exclude… regardless, of evolutionary interest" text in the revised approach to this discussion.

We thank the reviewer for this comment that was also pointed out by the other reviewers. To further assess the occurrence of independent gene duplications, we now expanded the retrotransposition analysis that we conducted in mouse and human genomes to pig, dog, and rat genomes. Notably, we found that lineage-specific L1 retrotransposons accompany amylase gene copy number gains in these species (see new Supplementary file 3and revised Figure 2B).This conceptual aspect of our approach is now described in the revised Results section and the technical details are provided in the main Materials and methods section. Overall, our analyses suggest that independent, lineage-specific gene duplications indeed occurred. However, we agree with the reviewer that we cannot fully exclude the role of other mechanisms at play in shaping the genetic variation in this locus. Thus, in our interpretation of these data in the Results and Discussion sections, we mentioned that additional mechanisms (e.g. gene conversion, incomplete lineage sorting, crossover events, and ancestral gene duplication polymorphisms) may also have contributed to the observed genetic variation in the amylase locus.

3) In general (as mentioned directly above, but this is actually a wide problem and a big issue) there is insufficient detail presented on the experimental results and analyses in the paper. While this issue can readily be addressed in revision, it means that reviewers may not be able to identify some problems from this version of the manuscript.

We apologize for the lack of detail in the previous version of our manuscript. We have now substantially revised both the Results and Materials and methods sections to include detailed information about experimental results and analyses. As we mentioned above, we also added a new Materials and methods section.

4) The finding of salivary amylase expression in some dog breeds is definitely interesting. However, I felt that the conclusions of AMY duplication 'bursts' necessarily being linked to the production of salivary amylase was too strong based on the data presented, and again seemingly not necessarily for this to be an interesting and valuable paper.

We agree with the reviewer. In fact, this same point was raised by the other reviewers as well. We found that all the species that express amylase in their saliva harbor at least one additional copy of the amylase gene. This could suggest that a gene duplication event may be a necessary first step for amylase to be expressed in salivary glands. As this reviewer correctly noted, we were not able to demonstrate a direct correlation between *AMY* gene copy numbers and enzymatic activity in saliva. We now revised our Results section to tone down our initially over-reaching conclusion.

5) Why are non-human primates analyzed separately from the non-primate mammal data?

We decided to dedicate a section to primates simply because we have better reference genomes, broader sampling, and accompanying dietary data for this branch of the mammalian tree. For example, we were able to detect more subtle changes, potentially driven by diet, in primates as compared to non-primate mammals in gene copy number and salivary expression (Figure 5). We felt that this warranted a separate section. We now clarified this reasoning at the beginning of the “Evolution of amylase in primates” section.

6) In the evolutionary analyses, some sort of approach that takes phylogenetic relationships into account should be used. Also, I understand the limitations with differentially available dietary starch intake data among species… however, could a subset of the dataset be formally analyzed in this framework, i.e. considering the species for which roughly equivalent dietary data are available?

The reviewer is right. To address this, we have now conducted an independent phylogenetic contrasts analysis (new Figure 4C). This analysis, which puts into consideration the phylogenetic distances, still showed that the trends we observed based on direct comparison remain robust. In addition, we used information found in the literature to update our analysis with the predicted starch consumption of the animals we tested. Specifically, we now included additional columns in Figures 4A and 4B to contrast high and low starch consuming species.

7) Details were insufficient for me to evaluate the AMY copy number simulation results, although it is difficult for me to imagine the parameters of the simulation exercise being able to accurately model mutation rates (and especially given that rates of NAHR once duplication has occurred are expected to be strikingly higher than the rate of initial duplication of a locus).

We have now added a detailed explanation of our modeling approach in the new Materials and methods. In addition, we have now updated our modeling approach with additional parameters to reflect expected mutational mechanisms. For example, we considered, as the reviewer suggested, that the mutation rate of a single copy state is expected to be significantly lower than at a state with two or more copies where nonallelic homology is established.

Reviewer #2:In this paper, employing digital PCR experiments to assess amylase copy number across a diverse range of mammalian lineages, Pajic et al. identify higher copy number within certain starch-consuming lineages. Additionally, they perform assays to measure salivary amylase activity and coalescent simulations to model neutral copy number variant evolution.This work is performed in the context of known amylase gene family expansions in human, rodents, and dog, so the novelty here is the comprehensive nature with which sampling and analysis took place. The data presented here is confirmatory of the previously known amylase gene family expansions without addressing some relevant previous work, notably positive selection identified to be acting on amylase loci in mouse and dog (Staubach et al., 2012, Reiter et al., 2016).

We thank the reviewer for directing us to these relevant publications. In our revised Introduction section, we now explicitly mention their findings describing potential adaptive evolution of the amylase locus in house mice and dogs.

The possibility of gene conversion and the effect this could have on the data and inferences presented here is not discussed. There is evidence of gene conversion within this locus in human (Groot et al., 1990) and only the scenario of numerous gene duplication events and one gene loss event is presented in the section 'Evolution of amylase in primates'. If gene conversion is taking place, then duplicates may appear to be younger and more lineage-specific. Alternative scenarios are not discussed as other explanations of the data observed. Fewer and older duplication events could have occurred, followed by more than one gene loss event and this history is masked by lineage-specific similarity as a consequence of gene conversion. Additionally, ancestral duplication polymorphisms, incomplete lineage sorting and gene conversion is a possibility that is not adequately dismissed with the data and arguments presented.

To provide more evidence for lineage-specific duplications in the *AMY* locus, we expanded our analyses of the lineage-specific retrotransposons in this locus in pigs, rats, and dogs as described in our response to reviewer #1 (revised Figure 2B). However, the reviewer is right in pointing out the potential roles of other mechanisms. In our revised manuscript, we now mention gene conversion, ancestral duplication polymorphisms, and incomplete lineage sorting, as additional factors in shaping variation in this locus.

The maximum likelihood phylogenetic tree constructed using amylase protein sequences translated from reference genomes (Figure 2A) shows species-specific clustering of sequences. The phylogenetic tree would be more informative if it also including sequences from species where increases in copy number have not occurred and was rooted with an outgroup, which are both readily available.

We thank the reviewer for this helpful advice. We now included outgroup species in our analysis as suggested by the reviewer. Since adding these species to our main Figure 2A would have made this tree too voluminous, we chose to show this larger tree in a new Figure 2—figure supplement 1instead. The results of this analysis did not change our conclusions based on the data presented in Figure 2A.

The authors state that "amylase gene copy numbers in species correlate well with observable enzymatic activity in saliva (Figure 3C)" and additionally state that "the copy annotated as mouse AMY1 (Figure 2) is expressed in salivary glands, and is likely responsible for salivary expression of amylase in mice, while the other amylase duplicates have a negligible expression in salivary gland tissue (Figure S2)." However, they do not adequately reconcile these two statements. If only one copy in mouse is meaningfully expressed in saliva, what does that mean for salivary expression correlation with copy number? All but one copy have no impact on salivary activity so, for mouse at least, a relationship between salivary expression level and number of copies is not direct. Perhaps salivary expression level is correlated with a starch-rich diet and a starch-rich diet is also correlated with pancreatic expression. Increased pancreatic expression could be facilitated through gene duplication and increased salivary expression through enhancers? Discussing these alternative means of achieving increased expression in different tissues given the data observed would be welcome.

We completely agree with the reviewer and apologize for not having discussed this aspect of our results adequately. As already stated in our above response to Reviewer #1 we found no species with salivary enzymatic activity that does not have an *AMY* gene duplication. This suggests that duplication may be a necessary first step towards salivary gland-specific expression. Overall, however, there is no linear correlation between *AMY* gene copy number and salivary expression. We now describe these findings more carefully and discuss them in the revised Results and Conclusion sections.

Reviewer #3:In "Amylase copy number analysis in several mammalian lineages reveals convergent adaptive bursts shaped by diet" Pajic et al. examine amylase copy number across the mammalian phylogeny. The authors observe that amylase copy number appears to be higher in species that have high-start diets than those that do not, and that this correlates with levels of salivary amylase. Although these are some compelling observations, the authors claims of "adaptive bursts" and convergence in the title are both unjustified in my opinion.Regarding "adaptive bursts": this is largely supported by the authors evidence of independent origins of amylase copies. However, the protein sequence tree shown in Figure 2 could be effected by interlocus gene conversion, which will cause copies in the same species to appear more closely related. Thus we cannot be certain of the extent of the "burstiness" of the duplication events.

The reviewer is right and echoes the other reviewers’ concerns. As described above in our responses to reviewer #1 and reviewer #2, we now expanded our analyses of the lineage-specific retrotransposons associated with amylase gene copies in this locus in pigs, rats, and dogs (revised Figure 2B). This analysis provided more evidence that gene copy number expansions occurred in a lineage-specific manner. Furthermore, we now discuss gene conversion as a potential additional factor shaping variation in this locus and toned down our argument with regards to the “bursts” of amylase copies.

The authors have also performed simulations of gene family evolution under various mutation rates, but I found this analysis rather simplistic (e.g. no difference in gain and loss rates allowed), qualitative (no model fits are formally assessed), and incomplete (no alternative models including natural selection are considered).

We agree with the reviewer. We now expanded the description of our simulation results and added additional mutation rates. We also allowed dynamic copy number gain and loss rates in our revised simulations (now discussed in detail in the new Materials and methods section). However, since we have insufficient knowledge of the adaptive forces that may have acted on the amylase locus, we refrained from simulating alternative adaptive scenarios. We also toned down our formerly perhaps too assertive statements throughout the manuscript, including in the Title and the Abstract.

On a related note, in Figure 5, it is not clear to me how the authors placed the locations of duplication events across the phylogeny, and how confident we can be in these placements regardless of whether gene conversion is taken into account. Overall I would say that while the authors' observations are interesting and highly suggestive, there is no direct evidence for positive selection in these data, and thus the language about adaptation sprinkled throughout the manuscript should be removed or rephrased.

We agree with the reviewer on both accounts. We now discuss the caveats of interpreting the data shown in Figure 5in the Results section. In addition, we toned down our arguments throughout the manuscript with regards to adaptive forces shaping the copy number state and enzymatic activity of amylase in saliva. As a consequence of that, we also removed the phrase “convergent adaptive bursts” from the title of our revised manuscript and replaced it with the word “independent”.

Second, the evidence for convergent evolution is limited. The strongest example of convergence that the authors give is in the third paragraph of the subsection “Recurrent amylase copy number gains in multiple mammalian lineages”, where they describe that in both humans and mice there are tandem arrays of amylase genes with a flanking transposable element, suggesting the possibility of repeat-induced expansion of this gene family in both species. However, in mouse these copies do not encode a salivary amylase according to Figure 2. Thus this example provides no support for convergence related to salivary digestion.

We thank the reviewer for raising these questions which made us rethink and better clarify our interpretation of the data. In addition, we were able to find specific retrotransposons flanking rat, pig, and dog amylase gene copies, thereby providing lineage-specific signatures that further support our hypothesis that *AMY* gene copy number expansions occurred independently in different mammalian lineages. We also agree that enzymatic activity of amylase in saliva is not directly correlated with the number of *AMY* gene copies. Nevertheless, it is important to note that in *all* the species analyzed, salivary expression of amylase *always* went along with an amylase gene duplication event. We never intended to claim that the retrotransposon insertions by themselves lead to the expression of a given amylase copy in salivary glands. We also acknowledge that we do not have direct evidence for convergence of the salivary expression trends and, thus, removed this argument from our revised manuscript altogether.

Third, the authors examine in Figures 3 and 4 evidence of the relationship between amylase copy number and salivary amylase activity. However as far as I can tell the authors have not assessed the strength of this relationship statistically in a manner that accounts for phylogenetic relatedness among data points. Moreover, to me the authors' results imply that the amylase copy numbers are cannot explain a sizable portion of the variation in amylase activity among mammals. This is demonstrated in Figure 4, where the authors show amylase copy number along a linear scale and salivary amylase activity in log scale. For amylase copy number the difference between the "specialized" and "broad range" diet classes spans about a factor of 2- or 3-fold. For salivary amylase, the difference is about 2-3 orders of magnitude. Assuming that copy number has at best a linear effect on protein dosage, this result strongly argues against copy number having a very important role in salivary amylase activity in mammals.

The reviewer is right. In fact, the same point was raised by the other reviewers as well. In our analyses, we found no species with salivary expression above the basal levels that do not also have an *AMY* gene duplication. This suggests that duplication may be a necessary step towards the gain of expression in salivary glands. Beyond that, however, as the reviewer pointed out, there is no correlation across species between the number of *AMY* gene copies and the enzymatic activity of amylase in saliva. We now revised the Results section to reflect these insights.

Reviewer #4:The authors seek to understand whether the evolution of additional AMY genes occurred independently in different mammalian lineages, and whether the driving force behind AMY gene expansion was a starch-rich diet. To do this, they use a multi-pronged approach to investigate their research questions, including enzymatic assays, ddPCR to measure AMY gene copy numbers, and estimating copy numbers from reference genomes. The resulting dataset is extensive, with data on salivary amylase expression and AMY copy number variation in a large sample of mammals, significantly expanding our current knowledge about the evolution of salivary and pancreatic amylase, as well as the link between digestive enzyme evolution and diet more broadly. The manuscript is well-written and the figures effectively communicate the findings.The AMY gene copy numbers are quite consistent between ddPCR and reference genome estimates, providing confidence in the authors' findings regarding copy number variation.Interestingly, the authors find salivary amylase expression in mammals that were previously found not to express amylase in their saliva, including New World primates and dogs. Given that previous research did not find evidence for salivary amylase expression in these species (Perry et al., 2007; Axelsson et al., 2013), I think some skepticism is warranted. The authors state that dogs, for example, express "substantial amounts" of amylase in their saliva, however, based on the data in Supplementary file 1, amylase activity was quite low in most dogs compared to humans and rodents. Similarly, amylase activity in capuchins, pigs, and boars seemed quite low compared to catarrhines and rodents, making me wonder if some of the amylase "activity" may have been contamination or a basic level of activity that is always present. Was a negative control used? If so, it would be helpful to include the activity measures obtained for the negative control in Supplementary file 1. If a negative control was not used, the authors should repeat the assays with one.

We agree with this reviewer in that our description of amylase activities was too simplistic and may have been prone to misunderstanding. As a control, we now measured the amylase enzymatic activity in serum of several species as a background value. We now include this information as a threshold value, depicted as a dotted line in revised Figure 3C. Thus, only those values of salivary amylase activity that range above this threshold are considered as positive. Based on this, there are indeed some dog breeds/samples, that have to be considered “negative” for salivary amylase. However, others range clearly above the cut-off line, implicating that expression of amylase in dog saliva is not universal and might have occurred only in certain breeds. To further aid in the visual interpretation of the data, we added images of the lysis diameters corresponding to salivary amylase activity in the starch lysis plate assay next to the corresponding activity values in the y-axis of revised Figure 3C.

In the Results, the authors state that salivary amylase activity varied across different dog breeds and refer the reader to Supplementary file 1. I was curious to see which dog breeds have higher salivary amylase activity, however, Supplementary file 1 does not include information on the different dog breeds included in the sample. I think this information should be included and the differences might be worthy of additional discussion. Do breeds associated with high-starch human populations have higher salivary amylase activity, for example?

We agree with the reviewer that this is an interesting observation and that the adaptation to starch-rich diets in dogs (similar to the observations in humans) is worthy of further investigation. We now included the dog breeds in our revised Supplementary file 1.Our sample size, however, does not allow to conclusively test the correlation between different breeds and their corresponding salivary amylase levels. Further, we believe that more recent intentional breeding may have confounded the effect of starch intake that might have originally affected salivary amylase expression at a time when dogs became followers of human groups. These questions warrant a separate, much larger study solely focusing on dogs. We now discuss these thoughts in the Results section.

[Editors' note: the author responses to the re-review follow.]

Summary:The reviewers commend your efforts to revise your manuscript in response to our original reviews. The manuscript has been re-worked substantially, to positive effect. The reviewers felt that your primary conclusion that there have been bursts of amylase gene duplication in multiple lineages independently was now supported more strongly. Overall, the combination of your careful molecular genetics work on this complex locus alongside the protein expression measurements across a diversity of mammals is impressive, providing a substantial new advance into the evolutionary biology and evolutionary ecology of the amylase gene family, with broader molecular biology insights related to gene duplication and tissue expression pattern processes. We do collectively share some concerns about the manuscript that we ask you to address thoroughly.

We thank the editors and the reviewers for their thorough revision of our manuscript and the constructive comments. We now reworked our conclusion section to add a working model of the amylase locus evolution, removed the simulation component from this manuscript, and added a comprehensive literature review on starch consumption of mammals in the Materials and methods section. We also added a new figure (Figure 6) illustrating our current working model for the evolution of the amylase locus to accompany the conclusion section. We have addressed all the comments and believe that our revised manuscript has now improved substantially. Below we provide a point-by-point response to both “Essential Revisions”, as well as to individual reviewer comments. Please note that for ease of assessing the changes, we have highlighted them in the manuscript with yellow color.

Essential revisions:1) Clarify your methods for grouping broad-range diet mammals into lower-starch and higher-starch categories. We sympathize with the challenges of obtaining equivalent food consumption data for all species, but greatly increased detail is necessary here. This analysis should be fully transparent and replicable.

To address this concern, we have now conducted a comprehensive literature review on the diets of the species that we use in constructing Figures 4A and 4B. We now included this review in the Materials and methods section of the manuscript under the “Categorization of starch consumption” subsection. In this subsection, we clearly laid out our reasoning as to how we placed the mammals into lower- and higher-starch consuming categories.

2) The simulations as presented are uncompelling. If the authors choose to maintain this component of the manuscript, then the simulation approach and mechanics, process of parameter selection (using a range of values encompassing extremes of reasonable possibility, with clear justifications), and results all need to be clarified and detailed much more extensively. Is the diet information included in the model (i.e. asking how often neutral processes can explain the observed pattern whereby species with higher-starch diets are much more likely to have duplication bursts)?

After considering the editors’ and reviewers’ comments, we agree that the simulation data at this point do not add much to our conclusions and need substantial improvement. Thus, we decided to remove the simulation component from the manuscript.

3) Explicitly present a more complete/clarified model (or models) for amylase duplication and the gain of salivary amylase that could produce the observations in the paper. Some thoughts along these lines: If the model is simply that duplication must occur prior to the acquisition of salivary expression, then this is essentially a model of neofunctionalization (here, convergent in multiple mammal lineages for which starch is a substantial part of the diet). This model would posit that a single amylase copy cannot do the job of pancreatic and salivary amylase, perhaps due to changes to the amino acid sequence required for proper amylase activity (this is unlikely?), or instead that something about the regulatory structure of amylase prevents expression of a single copy in both the pancreas and in salivary glands. Also, the authors' results and interpretation now implies that just one new gene copy is required for the acquisition of sufficient salivary amylase activity, so the presence of additional copies in some species should be explicitly addressed as part of this discussion. We recommend creating a clearly defined section (e.g. 'Model for amylase gene locus evolution' or similar) in which this discussion can be concisely presented (note that a few relevant existing statements along these lines can be moved and modified from other sections), without needing to be conclusive. Opportunities for future investigation can be noted.

This was an important suggestion for which we are grateful. We now renamed our conclusion section as “Conclusion and outlook: a working model explaining how the amylase locus evolved” and substantially revised and expanded this section to include a logical working model of evolution of the amylase locus. To support our explanation, we also included a new figure (Figure 6).